



# Temperature sensitivity of soil organic carbon respiration along the Rwenzori montane forests elevational transect in Uganda

Joseph Okello[1,2,3,4], Marijn Bauters[1,2], Hans Verbeeck[2], Samuel Bodé[1], John Kasenene[3], Astrid Françoys[1,5,6], Till Engelhardt[7], Klaus Butterbach-Bahl[8], Ralf Kiese[8] and Pascal Boeckx[1]

[1]Isotope Bioscience Laboratory – ISOFYS, Ghent University, Coupure Links 653, 9000 Gent, Belgium
[2]CAVElab- Computational and Applied Vegetation Ecology, Ghent University, Coupure Links 653, 9000 Gent, Belgium
[3]School of Agriculture and Environmental Sciences, Mountains of the Moon University, P.O Box 837, Fort Portal, Uganda
[4]National Agricultural Research Organisation, Mbarara Zonal Agricultural Research and Development Institute, P.O Box 389, Mbarara, Uganda
[5]Soil Fertility and Nutrient Management (SoFer), Ghent University, Coupure Links 653, 9000 Gent, Belgium
[6]Soil Physics (SoPHy), Ghent University, Coupure Links 653, 9000 Gent, Belgium.
[7]Sweco, Arenbergstraat 13, 1000 Brussels, Belgium
[8]Institute for Meteorology and Climate Research, Atmospheric Environmental Research (IMK-IFU), Karlsruhe Institute of Technology, Kreuzeckbahnstrasse 19, Garmisch-Partenkirchen 82467, Germany

*Correspondence to:* [Joseph.Okello@UGent.be](mailto:Joseph.Okello@UGent.be)

Key words: Temperature sensitivity, $CO_2$ respiration, climate warming, Afromontane forests



## Abstract

Tropical montane forest store high amounts of soil organic carbon. However, global warming may affect these carbon stocks
by enhancing soil organic carbon respiration. Better insight into temperature response of soil organic carbon respiration can
be obtained from *in* and *ex situ* warming studies. *In situ* warming via translocation of intact soil mesocosms was carried out
along an elevational transect ranging between ca. 1250 m a.s.l. in the Kibale Forest to ca. 3000 m a.s.l. in the Rwenzori
Mountains. Samples from the same transect were also warmed *ex situ* to determine temperature sensitivity. The *ex situ*
results revealed that along the natural climate gradient in the elevational transect, specific heterotrophic $CO_2$ respiration
decreased linearly by $1.01 \pm 0.12$ µg C h$^{-1}$ g$^{-1}$ SOC per 100 m of elevation increase. Similarly, the temperature sensitivity
increased from $1.50 \pm 0.13$ in the lowest elevation clusters to $2.68 \pm 0.25$ in the highest elevation cluster, showing a linear
decrease of $0.09 \pm 0.03$ per 100 m of elevation increase. Additionally, the $^{13}$C depletion factor of the respired $CO_2$ decreased
linearly by $0.23 \pm 0.04$ ‰ per 100 m of elevation increase. The results indicate an increased recalcitrance and decreased
mineralisation of soil organic carbon with elevation driven by decreasing soil temperature and pH. Subsequently, after two
years of *in situ* warming (0.9 to 2.8 °C), specific heterotrophic soil organic carbon respiration tends to be lower for warmed
as compared to control soil. Further, in warmed soils, $^{13}$C and content of soil organic carbon relatively increased and
decreased, respectively. This indicates increased mineralisation and depletion of readily available carbon during two years of
warming. In conclusion, our results suggest that climate warming may trigger enhanced losses of soil organic carbon from
tropical montane forests, due to a combination of a higher temperature sensitivity of mineralisation and soil organic carbon
content.



# 1 Introduction

Tropical forests store 55 % of the global forest carbon (C) stocks of which 56 % is stored in biomass, 32 % in soil and 12 % in litter and deadwood (Pan et al., 2011). These forests account for more than one-third of primary productivity (Beer et al., 2010; Pan et al., 2011), despite covering only less than 10 % of the global land area (Cuni Sanchez et al., 2021; Erb et al.,

2018). Their key role in the global C cycle is further demonstrated by the fact that tropical forests exchange more carbon dioxide ($CO_2$) with the atmosphere than any other ecosystem (Friedlingstein et al., 2020; Lewis et al., 2015; Singh, 2018), in part owing to their high C turnover rates (Sayer et al., 2011). Currently, tropical ecosystems are being subjected to global change, with detrimental consequences for its ecosystem services (Gütlein et al., 2018). The global temperature increase is being driven by global climate warming and local land use change (Friedlingstein et al., 2020; Zhang et al., 2005; Ipcc,

2018). For instance, worldwide, the average surface temperature has been rising consistently by between 0.95-1.20 °C from 1850 to 2020 (Ipcc, 2021). In addition to the global climate warming effect, the tropical forests are also experiencing local and regional temperature increases driven by land use change, which alters the fluxes of solar and thermal infrared radiation, sensible, and latent heat and ultimately changes the surface albedo (Mahmood et al., 2014; Zeng et al., 2021). As an example, between the year 2000 and 2014, agricultural expansion at the expense of montane forests caused a general

increase in air temperature of 0.05 ± 0.01 °C in the Albertine rift mountains of Africa, and air warming of up to 2 °C can occur under extensive deforestation (Zeng et al., 2021). The increase in global and regional temperature has the potential to drastically alter the C cycle in tropical forests (Mohan, 2019; Nottingham et al., 2020; Sayer et al., 2019), and may potentially drive accelerated soil organic C losses (Friedlingstein et al., 2020; Kim et al., 2016). However, the response of tropical forest soils to such temperature increases is largely uncertain, especially in mountainous areas (Nottingham et al.,

2020; Sayer et al., 2019; Zeng et al., 2021), and generally data on climate warming from tropical forests in Africa are hardly available.

Key in this ecosystem-level uncertainty, is the specific effect of warming on soil respiration rates (Carey et al., 2016; Crowther et al., 2016; Fussmann et al., 2014). Indeed, up to about 40 % of the $CO_2$ emissions from tropical forest ecosystems

originate from soil respiration (both autotrophic and heterotrophic respiration) (Malhi, 2012), of which approximately 60 % of the $CO_2$ respiration from soil is derived from heterotrophic microbial activity during the mineralisation of organic C (Sayer and Tanner, 2010). Temperature increases in tropical forests can trigger accelerated respiration rates, yet the ecosystems' net primary productivity is already close to its maximum, thereby reducing the net C sink (Carey et al., 2016; Craine et al., 2010; Nottingham et al., 2020). Therefore, given the sheer magnitude of both the C storage and emission

capacities of tropical forest soils, precise quantifications and assessment of their response to climate warming are needed to inspire climate-sensitive forest management and to improve parameterisation and predictions of earth system models (Oertel et al., 2016; Quéré et al., 2018).



Here we present new data from an eastern Afrotropical elevational transect, set up in the Kibale Forest and the Rwenzori

Mountains in western Uganda. To gain better insight into the drivers of soil organic C (SOC) respiration, in function of
elevation and enhanced warming, we investigated: (i) soil physicochemical properties and the microbial community
compositions based on phospholipid fatty acid analysis (PLFA), (ii) heterotrophic soil $CO_2$ respiration rate from laboratory
incubations at 60 % water-filled pore space (WFPS) and controlled corresponding *in situ* temperature, (iii) changes in
heterotrophic soil $CO_2$ respiration from intact mesocosm translocated *in situ* along an elevation gradient to simulate a

warming of about 2°C, (iv) activation energies (AE) and temperature sensitivities ($Q_{10}$) of heterotrophic soil $CO_2$ respiration
rates and finally, (v) seasonal total soil $CO_2$ respiration rate under *in situ* conditions. In particular, we intend to address the
following research questions, with respect to a tropical Afromontane elevational transect.

  i.  Does soil organic matter recalcitrance increase with elevation?

  ii.  How does soil organic carbon respiration respond to two years of *in situ* soil warming?


## 2 Materials and Methods

### 2.1 Study area

The study was conducted in the Kibale Forest National Park and the Rwenzori Mountains National Park in Uganda, both
protected by the Uganda Wildlife Authority. A total of twenty sampling plots, each measuring 40 m by 40 m, were

established between 1250 to 3000 meters above sea level (m a.s.l.) along an elevational transect. The sampling plots were
grouped into five elevation clusters, with each cluster consisting of four replicated plots within similar elevation and
environmental conditions. From the twenty sampling plots, four plots are located in the Kibale Forest National Park at an
elevation of 1250-1300 m a.s.l. to form the "premontane" elevation cluster. Sixteen sample plots are located at four different
elevation clusters (1750-1850, 2100-2200, 2500-2600 and 2700-3000 m a.s.l.) in the eastern slope of the Rwenzori

Mountains National Park (Figure 1).

The Kibale Forest National Park (795 km²; 00°30'N 30°24'E) is located in the Kabarole and Kamwenge districts of western
Uganda. The climate is moist tropical, and temperatures stay nearly constant all-year-round. The average annual rainfall is
1365 ± 53 mm and the average temperature is 27.8 ± 0.74 °C (data 1992-2012, Kyembogo weather station in Kabarole

district 20 km from the park, at elevation of 1400 m a.s.l., Ministry of Water and Environment). The dominant soil type
according to the World Reference Base (WRB) classification is Ferralsol (Jacobs et al., 2016).

The Rwenzori Mountains National Park (998 km²; between 0°06'S – 0°46'N and 29°47'E – 30°11'E) is located at the border
between the Democratic Republic of the Congo (DRC) and Uganda. The region experiences a moist tropical climate, locally
affected by altitude and topography. Annual rainfall varies with elevation and slope aspect, with the highest rainfall amounts





on the eastern slope, where our transect was established. Recent rainfall data from the Uganda Wildlife Authority from 2012
to 2015 showed variations in mean annual rainfall ranging from 7000 mm at 1760 m a.s.l. to 1570 mm at 4230 m a.s.l.. The
mean annual soil temperature of the different elevation clusters is indicated in Figure 1.The dominant soil type in the
Rwenzori Mountains according to WRB classification is Leptosol (Jacobs et al., 2016).

**2.2 Soil physicochemical properties and microbial community structure**

In each study plot, the soil temperature at 5 cm depth was measured daily (during the measurements of soil respiration) at an
interval of 30 minutes, using thermocron iButton sensors DS1921G-F5 (iButton, Thermocron Baulkham Hills, Australia).
Similarly, the daily volumetric soil moisture content was measured at 5 cm soil depth using soil moisture sensors (EC-5,
Decagon Devices, Armidale, Australia). Further, soil bulk density was determined using the soil core method (Campbell and
Henshall, 2000).


At each of the twenty sampling plots along the transect, four topsoil samples (0-10 cm) of 385 cm$^3$ by volume were collected
(i.e. one sample per 20 m by 20 m subplot within the 40 by 40 m sample plot) and homogenised to form one composite
sample per plot (4 replicate composite samples per elevation cluster). The samples were oven-dried at 60 °C and sieved
through 2 mm mesh size and ground. Subsequently, C, nitrogen (N) content and $\delta^{13}$C were determined from the composite

soil samples using an elemental analyser (automated nitrogen carbon analyser; ANCA-SL, SerCon, Cheshire, U.K.), coupled
to an isotope ratios mass spectrometer (IRMS; 20-22, SerCon, Cheshire, U.K.). To measure the soil pH, 5 mL of the oven-
dried soil (in triplicate) was brought into suspension with 25 ml of 1$M$ KCl (1:5 v/v) and shaken end-over-end for one hour.
Subsequently the suspension was left to settle for two hours, then soil pH was measured in the supernatant using a pH glass
electrode, (model 920A, Orion, England).


For determination of the microbial community structure, about 20 g of the homogenised composite sample from each plot
was frozen immediately after collection. The microbial community structure was determined using phospholipid fatty acid
(PLFA) analysis. The PLFA analysis was done by extracting 5 g of freeze-dried soil sample in duplicate following the
method described by Bligh and Dyer (1959), and as modified by Findlay et al. (1989). Briefly, the method involves

extraction of all fatty acids, followed by isolation of phospholipids from other soil lipids (using solid-phase extraction), and
finally the conversion into fatty acid methyl esters. Accordingly, for each gram of soil sample, lipid extraction was done
using a combination of 0.1$M$ phosphate-buffer, trichloromethane and methanol solvents (0.9:1:2, v/v) at 25°C. Subsequently,
the volume of the total lipid extracts was reduced by evaporating the solvent from tubes under N gas in a water bath at 30 °C.
After, the neutral lipids and glycolipids were eluded using trichloromethane and acetone respectively. Phospholipids were

further eluded using methanol and concentrated by evaporation of the solvent under N gas in a water bath at 30 °C.



Eventually, the phospholipid fatty acids were converted to methyl esters, which were subsequently analysed using gas chromatography (GC, Trace GC, Thermo Scientific, Bremen, Germany).

## 2.3 Laboratory incubations

To assess soil heterotrophic $CO_2$ respiration rates under controlled laboratory incubations, the homogenised composite samples from each plot were air-dried and sieved (2 mm mesh size) to remove coarse particles and roots. For the incubation experiments, 50 g of each air-dried composite soil sample from each plot (4 replicates per elevation cluster) was placed in a gas jar of 1 L, which could be closed in an air-tight way by a lid. To each sample, deionised water was added until 60 % WFPS of the respective soil sample (based on total porosity derived from bulk density measurements), representing a moisture content for optimal microbial activity (Aon et al., 2001; Doetterl et al., 2015). The samples were then pre-incubated

for 14 days (at the respective *in situ* mean annual temperature per elevation cluster, i.e. 20, 17, 15, 13 and 12 °C for elevation clusters of 1250-1300, 1750-1850, 2100-2200, 2500-2600 and 2700-3000 m a.s.l. respectively). During the pre-incubation, the gas jars were closed with parafilm to permit free air circulation while minimizing the loss of water.

After 14 days of pre-incubation, each sample was removed from the incubator and flushed for 10 seconds with ambient air in

the room by means of an air fan. Immediately, one gas sample was taken using a 45 mL syringe and the ambient $CO_2$ concentrations, and $\delta^{13}C$ isotopic composition of ambient $CO_2$ at "open conditions" analysed using Cavity Ring-Down Spectrometer, (G2113-I, CRDS $CO_2$ analyser, Picarro, United States) at starting condition. The jars were then closed and placed back in the incubators for 24 hours (a preliminary trial experiment indicated a continuous linear increase in headspace $CO_2$ concentrations during 24 hours). After 24 hours, a 45 mL gas sample was taken and was immediately introduced into

the cavity ring-down spectrometer for measurement of the $CO_2$ concentration and $\delta^{13}C$ isotopic composition of the respired $CO_2$. After collecting the gas samples under closed conditions, the jars were opened, soil moisture replenished to 60 % WFPS, after which the jars were covered with parafilm and placed back in the incubator until the following measurement (to avoid gas accumulation and to re-establish ambient $CO_2$ concentrations). The following day, the same procedure was repeated, and this was done for five consecutive days to attain five replicated $CO_2$ concentrations that were used to calculate

the average respiration rate. To determine the $\delta^{13}C$ of the respired $CO_2$, we used the Keeling mass balance approach (Keeling, 1958) (equation 1).

$$\delta^{13}C\text{-}CO_2 = \frac{F^*f - I^*i}{F - I} \tag{1}$$

Where:

F = final concentration of $CO_2$ in the headspace;

f = final $\delta^{13}C$ of $CO_2$ in the headspace;

I = initial concentration of $CO_2$ in the headspace;





i = initial $\delta^{13}C$ of $CO_2$ in the headspace.

Subsequently, the isotopic depletion factor, epsilon ($\varepsilon$), i.e. the extent to which the product of respiration (i.e. $CO_2$) becomes depleted in $^{13}C$ during SOC (substrate) respiration was determined using equation 2.

$$\varepsilon = \left( \left( \frac{1000 + \delta^{13}C\text{-SOC}}{1000 + \delta^{13}C\text{-}CO_2} \right) - 1 \right) * 1000 \qquad (2)$$

Where:

$\varepsilon$ is the isotopic depletion factor

$\delta^{13}C$-SOC is the $\delta^{13}C$ of the soil organic carbon

$\delta^{13}C\text{-}CO_2$ is the $\delta^{13}C$ of the respired $CO_2$

The component $\left( \frac{1000 + \delta^{13}C\text{-SOC}}{1000 + \delta^{13}C\text{-}CO_2} \right)$ is alpha ($\alpha$), which is the isotopic fractionation factor.

### 2.4 Long-term *in situ* warming: a soil mesocosm translocation along elevational transect

*In situ* climate warming was simulated by translocating intact soil cores (16 cm diameter and 25 cm depth increment) along the altitudinal gradient to the nearest elevation cluster downslope (Figure 1). These intact soil cores were taken using a metallic soil corer in which a plastic PVC tube was inserted to collect an intact soil mesocosm. From each plot, four soil
mesocosms were translocated downslope (hereafter referred as "warmed"), while four mesocosms were transplanted within the same plot (hereafter referred as "control"). Each elevation cluster (except the highest) therefore had a total of 16 warmed and 16 control soil mesocosms. The soil mesocosms that were translocated from higher to lower elevation clusters were warmed by about 0.9 to 2.8 °C on average for two years (Figure 1).





Figure 1. The height profile of the Rwenzori elevational transect starting from nearby premontane Kibale Forest National Park (1250-1300 m a.s.l.) onto the Rwenzori Mountains National Park (1750-3000 m a.s.l.). Mean annual *in situ* temperatures for each of the five elevation clusters are indicated. The set-up of *in situ* warming through downslope translocation of soil cores to the immediate lower elevation cluster is illustrated on the left side of the scheme. Different colours on the elevation axis represent the colour code for each elevation cluster throughout the manuscript.



After 650 days of *in situ* incubation between November 2017 to September 2019 of both control and warmed soil cores, the cores were collected. The top 10 cm of the soil cores was collected (i.e. the soil layer with the highest C content and most active in C cycling), homogenised, air-dried, and sieved (2 mm mesh size) for additional laboratory incubation experiments, in order to assess the effect of two years of *in situ* warming on: (i) $CO_2$ respiration rates, (ii) the AE and $Q_{10}$ coefficient, (iii)
SOC and soil $\delta^{13}C$ isotopic composition.

To assess the above parameters (i-iii), the soil samples from the translocation experiment were subjected to another incubation experiment under optimal soil moisture conditions. Firstly, the control and warmed samples were incubated at the corresponding mean annual soil temperatures at which they were transplanted *in situ* (Table 1). Additionally, subsamples
from the same soil were also incubated at four other temperatures (5, 10, 15 and 30 °C, covering the temperature ranges in the entire elevational transect) to allow determination of AE and $Q_{10}$ through curve fitting of the $CO_2$ respiration rates at five different temperatures.

For these experiments, 6-15 g of air-dried soil (where the mass depended on the bulk density) was placed in the gas jar (50
mL by volume) for each temperature treatment and moistened with deionised water to attain 60 % WFPS. The soil was then gently compressed to a pre-determined height in correspondence with the bulk density of the undisturbed soil. Subsequently, the jars were weighted, covered with parafilm and pre-incubated for 14 days to allow for the re-activation and stabilisation of microbial activities. During the pre-incubation, the soil moisture content in each jar was monitored, and when needed corrected by adding deionised water. After 14 days of pre-incubation, the collection of air samples for the determination of
$CO_2$ respiration rate was initiated by removing the parafilm, aerating the samples and closing the lids in an air-tight way. The $CO_2$ concentrations at starting condition and after 24 hours were analysed using a gas chromatograph (Finnigan Trace GC Ultra, Thermo Electron Corporation, Milan, Italy) fitted with a thermal conductivity detector. Eventually, the $CO_2$ respiration rate was determined as the slope of $CO_2$ concentration in function of time.

### 2.5 *In situ* total soil $CO_2$ respiration

Along the Rwenzori elevational transect, we selected one sampling plot in each of the four elevation clusters at 1250-1300, 1750-1850, 2100-2200 and 2700-3000 m a.s.l., for the collection of *in situ* gas samples. In each study plot, *in situ* total soil respiration rates were measured following the static chamber method (Collier et al., 2014). Each sampling plot was subdivided into four 20 m by 20 m subplots. In each plot, five positions for static chambers were identified, with one position per 20 m by 20 m subplot, and another one in the center of the 40 m by 40 m permanent sample plot. The collars of
the static chambers (to anchor the chamber in the soil) were installed in the soil at least 24 hours before the first gas sampling event and were maintained in the field throughout the sample collection periods to minimise the effect of soil disturbance. Natural litter cover was left intact but plants were not included within the measuring chamber. For respiration rate



measurements a static, opaque gas chamber was placed on top of the collar, to create a headspace in which $CO_2$ emitted from the soil can accumulate. Both the collars and the chambers are made of polyvinylchloride (PVC) material and were painted

white to limit heating of the chamber's headspace air. The chamber was equipped with a vent tube to minimise pressure differences, and a septum connected to a three-way valve to allow the collection of headspace air samples. The headspace volume of the static gas chamber was 5 L and the surface area was 0.019 $m^2$.

During each sampling event, the five gas chambers were closed for 90 minutes, during which 15 mL air samples were

collected with a syringe from the chamber headspace at 30-minute intervals starting at time 0 minute until 90 minutes (i.e. $t_1$ = 0, $t_2$ = 30, $t_3$ = 60, and $t_4$ = 90 minutes). Prior to air sample collection, the gas chamber headspace was flushed three times with its headspace air using the sampling syringe to homogenise the air in the gas chamber headspace. Headspace air samples of 15 mL each were immediately injected in 12 mL pre-evacuated air-tight vials (Labco, Lampeter, Wales, U.K) that were closed with a silicone septum (Dow Corning 734). This created a slight over-pressure in the vials. The headspace

air samples were collected for five consecutive days during the start of the rainy season (August 2019), and the same process was repeated in the mid rainy season (September 2019) to account for any seasonal variations in environmental conditions. $CO_2$ emission measurements were not done in the dry season because microbial respiration and temperature sensitivity are low when the WFPS is below 30 % (Aon et al., 2001). The collection of headspace air samples was always consistently executed between 11:00 and 13:00 hours in all plots to minimize the effect of diurnal temperature differences (Keane and

Ineson, 2017).

## 2.6 Determination of $CO_2$ respiration rates, AE and $Q_{10}$

To convert the measured $CO_2$ concentrations into respiration rates, we fitted a linear regression of the concentrations over time. The derived soil $CO_2$ respiration rates were then expressed in units of grams of C per unit area per hour (for *in situ* measurements) or units of grams of C per unit of soil per hour or normalised per unit SOC (for laboratory incubations) to

obtain the "specific" heterotrophic $CO_2$ respiration rate. This was done using the ideal gas law as described in equation (3) to obtain the net gas respiration rates taking into account the headspace volume of the gas chamber, pressure, temperature and molar weight of the gas (Collier et al., 2014; Dalal et al., 2008; Kutzbach et al., 2007).

$$F_C = \left[\frac{\Delta C}{\Delta t}\right] * \left[\frac{P*V}{R*T*A}\right] * M_w \tag{3}$$

Where:

$F_C$ is the resulting gas respiration rate in (g C $m^{-2}$ $h^{-1}$) (for *in situ* respiration rates)

$\Delta C$ is the change in gas concentrations (ppm)

$\Delta t$ is the change in incubation time (hour)

P is the pressure (atm)





V is the volume of the gas chamber headspace (L)

R is the molar gas constant (L atm mol$^{-1}$ K$^{-1}$)

T is the absolute temperature (K)

A is the surface area of the gas chamber (m$^2$)

$M_w$ is the molar weight (g mol$^{-1}$)

For the laboratory incubation experiments, the parameter A (i.e. the surface area of the gas chamber) was replaced either by the weight of the incubated soil (to express it as "mg C h$^{-1}$ kg$^{-1}$ soil" or by the concentration of soil SOC (to express it as "µg C h$^{-1}$ g$^{-1}$ SOC").

To determine the activation energy of $CO_2$ respiration rate, we employed equation (4) with the respiration rate expressed per unit SOC.

$$F_C = b * e^{\frac{-AE}{R*T}} \tag{4}$$

Where:

$F_C$ is the specific $CO_2$ respiration rate (µg C h$^{-1}$ g$^{-1}$ SOC)

b is a pre-exponential factor (that is, the theoretical reaction rate constant in the absence of activation energy)

AE is the activation energy in kJ mol$^{-1}$

After log transformation, equation (4) becomes:

$$\ln F_C = AE * \left[\frac{-1}{R*T}\right] + \ln b \tag{5}$$

Hence, when plotting $\ln F_C$ against $\frac{-1}{R*T}$, the activation energy can be determined as the slope of the linear regression (SI, Figure 1).

To determine the coefficient for temperature sensivity of SOC respiration ($Q_{10}$), the $CO_2$ respiration at five different incubation temperatures were firstly fitted to an exponential function, i.e. equation (6) (SI, Figure 2).

$$F_C = a * e^{k*T} \tag{6}$$

From the two exponential regression constants a and k, the k value was used to calculate the $Q_{10}$, using equation (7).

$$Q_{10} = e^{10*k} \tag{7}$$





### 2.7 Data Analysis

To determine whether there was a difference in the $CO_2$ respiration rates, AE and $Q_{10}$ among the elevation clusters in the
elevational transect, we employed analysis of variance (ANOVA) to check differences in means of each variable. Where a
significant difference was detected, post-hoc analysis for multiple comparisons was performed using Tukey honest
comparison of means to explicitly reveal which elevation clusters differed from each other. We used quantile-quantile and
residual plots to check whether the data followed assumptions of ANOVA. To check whether there was a dependency of
$CO_2$ respiration rates, AE and $Q_{10}$ on elevation, we used the linear mixed effect model regression "lme4" package in R
software, in which elevation was used as fixed effect and elevation cluster location as random effect (to control for spatial
clustering of the sampling plots). To estimate the *P*-values, we used type III analysis of variance with Satterthwaite's
approximation method in the linear mixed effect model. In each linear mixed effect model, both marginal R square ($R^2_m$) and
conditional R square ($R^2_c$) values were obtained following Nakagawa and Schielzeth (2013). Further, to check for a change
in $CO_2$ respiration rates, AE, $Q_{10}$, SOC content and $\delta^{13}C$ isotopic composition between control and *in situ* warmed soil at
each elevation cluster, we used a Wilcoxon test. Subsequently, to check the effect of warming along the entire elevational
transect, we fitted linear mixed effect model for both control and warmed soil, where elevation was used as fixed effect and
the elevation clusters as random. Finally, we employed principal component analysis to explore changes in microbial
community and soil physicochemical properties along the elevational transect. All data were analysed using R software (R
Core Team, 2021), and a *P*-value of 0.05 was taken as significance level.



## 3 Results

### 3.1 Physicochemical soil properties and microbial community

The physicochemical soil properties in the Rwenzori elevational transect are described in





Table 1. Along the elevational transect, average annual soil temperature, bulk density and pH$_{KCl}$ decreased linearly. On the other hand, SOC, soil total N and carbon-to-nitrogen (C:N) ratio, increased linearly with increasing elevation. On the other hand, the δ$^{13}$C of the SOC showed no linear trend along the elevational transect.





**Table 1.** Physicochemical soil properties (0-10 cm) of the elevational transect on the East facing slope of the Rwenzori Mountains National Park (1750-3000 m a.s.l.) and the premontane Kibale Forest National Park (1250-1300 m a.s.l.). Indicated are the mean values plus/minus standard deviations of average annual soil temperature at 5 cm depth, bulk density ($\rho_b$), pH in KCl solution (pH$_{KCl}$), carbon (C) content, nitrogen (N) content, carbon-to-nitrogen ratio (C:N), and $\delta^{13}$C. The elevational trend from the linear mixed effect regression model estimate per 100 m of elevation increase is also indicated with the standard error (SE), *P*-value, marginal $R^2$ ($R^2_m$) and conditional $R^2$ ($R^2_c$).

| Elevation cluster (m a.s.l.) | Soil class | Soil temperature (°C) | $\rho_b$ (g cm$^{-3}$) | pH$_{KCl}$ | C (%) |
|---|---|---|---|---|---|
| 1250-1300 | Ferralsol | 19.5 ± 1.1[a] | 1.04 ± 0.09[a] | 5.41 ± 0.29[a] | 4.04 ± 1.41[c] |
| 1750-1850 | Leptosol | 16.7 ± 0.8[b] | 0.77 ± 0.13[b] | 4.13 ± 0.11[b] | 5.31 ± 2.15[c] |
| 2100-2200 | Leptosol | 15.3 ± 1.0[c] | 0.51 ± 0.07[c] | 4.00 ± 0.13[b] | 13.36 ± 0.68[b] |
| 2500-2600 | Leptosol | 13.0 ± 0.9[d] | 0.45 ± 0.07[c] | 3.56 ± 0.13[c] | 18.27 ± 3.58[a] |
| 2700-3000 | Leptosol | 12.1 ± 1.1[e] | 0.44 ± 0.08[c] | 3.25 ± 0.21[c] | 22.97 ± 4.29[a] |
| Fixed effect estimate per 100 m | NA | -0.51 | -0.04 | -0.13 | 1.33 |
| SE | | 0.00 | 0.00 | 0.02 | 0.22 |
| *P*-value | | <0.001* | 0.009* | 0.011* | 0.007* |
| $R^2_m$ | | 0.90 | 0.78 | 0.83 | 0.81 |
| $R^2_c$ | | 0.90 | 0.85 | 0.91 | 0.89 |





| | | | | | | | | | | |
|---|---|---|---|---|---|---|---|---|---|---|
| N (%) | $0.44 \pm 0.12^b$ | $0.60 \pm 0.24^c$ | $1.09 \pm 0.17^b$ | $1.40 \pm 0.20^{ab}$ | $1.81 \pm 0.25^a$ | 0.09 | 0.01 | 0.006* | 0.82 | 0.87 |
| C:N | $9.1 \pm 0.7^b$ | $8.9 \pm 0.5^b$ | $12.5 \pm 1.6^a$ | $13.0 \pm 0.7^a$ | $12.6 \pm 0.9^a$ | 0.31 | 0.09 | 0.041* | 0.59 | 0.82 |
| $\delta^{13}C$ (‰) | $-25.5 \pm 0.6^{ab}$ | $-24.5 \pm 0.9^a$ | $-25.3 \pm 0.6^{ab}$ | $-25.8 \pm 0.7^{ab}$ | $-26.4 \pm 0.3^c$ | -0.06 | 0.05 | 0.345 | 0.13 | 0.51 |

Different lowercase letters in superscript (bold) next to values of each elevation cluster (same row) indicate a significant difference among the sites at $P < 0.05$. The mean values were calculated from 4 separate composite soil samples per elevation cluster and are expressed per unit of dry soil. The $P$-values for a statistically significant elevational linear trends are

marked with an asterisk symbol "*".





Further, at the start of the rainy season, the average *in situ* soil temperature at 5 cm depth decreased from $19.6 \pm 0.3$ °C at 1250-1300 m a.s.l. to $11.9 \pm 0.5$ °C at 2700-3000 m a.s.l. (SI, Table 3). Generally, along the elevational transect, the average soil temperature at the start of the rainy season decreased linearly at a rate of $0.50 \pm 0.04$ °C per 100 m of elevation increase ($R^2_m = 0.96$, $P = 0.006$, SI, Figure 3). On the other hand, in the mid rainy season, the average *in situ* soil temperature

decreased from $20.4 \pm 0.4$ °C at 1250-1300 m a.s.l. to $12.3 \pm 0.4$ °C at 2700-3000 m a.s.l.. Similarly, in the mid rainy season, the soil temperature decreased linearly by $0.52 \pm 0.06$ °C per 100 m of elevation increase ($R^2_m = 0.92$, $P = 0.014$, SI, Figure 3). Additionally, at each elevation cluster, the average soil temperature at 5 cm depth was significantly higher in the mid rainy season than at the start of the rainy season (SI, Table 3, SI, Figure 3).

In addition, at the start of the rainy season, the average percentage WFPS was $33.1 \pm 1.2$ % at 1250-1300 m a.s.l., $22.2 \pm 2.5$ % at 1750-1850 m a.s.l., $41.7 \pm 2.9$ % at 2100-2200 m a.s.l., and $42.7 \pm 8.7$ % at 2700-3000 m a.s.l. (SI, Table 3). Generally, along the elevational transect, the average WFPS showed no significant elevational trend at the start of the rainy season (SI, Figure 3). In the mid rainy season, the average percentage water-filled pore space was $57.2 \pm 5.8$ % at 1250-1300 m a.s.l., $44.8 \pm 3.8$ % at 1750-1850 m a.s.l., $45.4 \pm 4.8$ % at 2100-2200 m a.s.l., and finally $44.5 \pm 9.0$ % at the highest elevation

cluster (2700-3000 m a.s.l.) (SI, Table 3). Generally, along the elevational transect, the average WFPS showed no significant elevational trend in the mid rainy season (SI, Figure 3). On the other hand, the soil moisture content was always higher in the mid rainy season in all elevation clusters except at 2700-3000 m a.s.l. (no significant difference) (SI, Table 3).

Finally, along the elevational transect, the microbial community composition showed no significant trend (SI, Figure 4). The

percentage of the variability explained by elevation was low for each microbial group, i.e. gram-positive bacteria ($R_m^2 = 0.07$), gram negative bacteria ($R_m^2 = 0.01$), fungi ($R_m^2 = 012$), total PLFA (bacteria plus fungi) ($R_m^2 = 0.06$), the ratio of gram-positive to gram-negative bacteria ($R_m^2 = 0.05$) and the ratio of bacteria to fungi ($R_m^2 = 0.10$) (SI, Figure 4). Subsequently, the principle component analysis (PCA) of soil parameters (including microbial community composition), further confirmed that the ratio of bacteria to fungi depicted a weak negative correlation with elevation. Further, the ratio of

gram-positive to gram-negative bacteria revealed a weak positive correlation with elevation (Figure 4, SI, Table 2). The parameters of principle component 1 with correlation scores above 0.5 included the total PLFA (bacteria plus fungi), fungi, gram-positive and gram-negative bacteria and physicochemical soil properties (soil bulk density, temperature, pH, SOC, total N, C:N). On the other hand, parameters of principle component 2 included the total PLFA (bacteria plus fungi), fungi, gram-positive and gram-negative bacteria, and the ratio of gram-positive to gram-negative bacteria and soil pH. Generally,

the PCA indeed revealed that the five elevation clusters have similar soil physicochemical properties. Further, it shows that the microbial community vectors are roughly orthogonal to the vectors of soil physicochemical properties. Up to 57.0 % and 23.3 % of the variability in the parameters were explained by principle component 1 and 2, respectively (Figure 2, SI, Table 2).



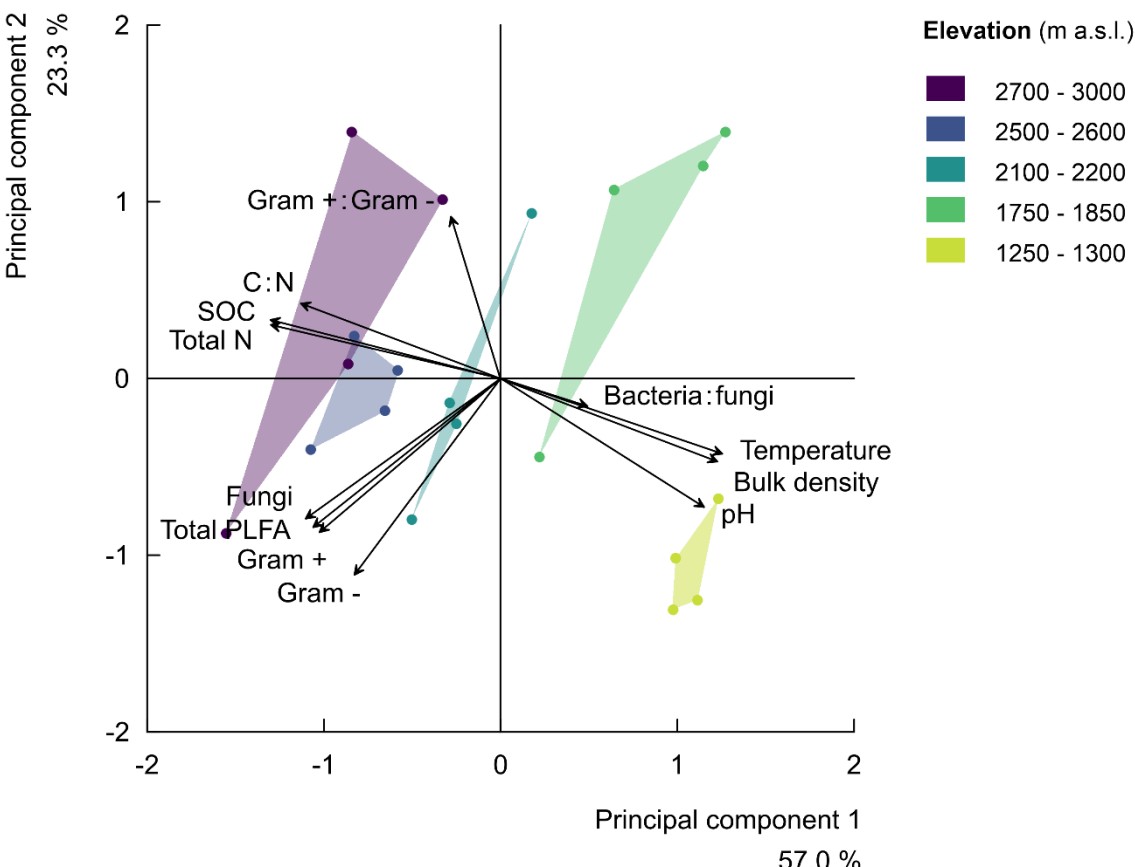

**Figure 2. Principal component analysis (rescaled) of the relationships along elevational transect for soil physicochemical properties (i.e. the soil total nitrogen (total N), soil organic carbon (SOC), carbon-to-nitrogen ratio (C:N), soil pH (pH), soil temperature at 5 cm depth (temperature), bulk density) and the microbial community compositions (i.e. the bacteria to fungi ratio (bacteria:fungi) (total PLFA, in nmol g⁻¹ soil), the ratio of gram-positive-to-gram-negative bacteria (gram+:gram-). The principal component 1 explained 57.0 % of the variability in studied parameters, while principal component 2 explained 23.3 % of the variability. The symbols of the individual plots on the biplot of the first two principal components are shown in different colours per elevation cluster.**



### 3.2 Laboratory-based heterotrophic soil $CO_2$ respiration rate, AE and $Q_{10}$

The heterotrophic $CO_2$ respiration rates from laboratory incubations at corresponding mean *in situ* temperature only showed a significant difference when comparing elevation cluster at 2100-2200 m a.s.l. ($CO_2$ respiration rate of $0.71 \pm 0.27$ mg C h$^{-1}$
kg$^{-1}$) with the elevation cluster at 2700-3000 m a.s.l. ($CO_2$ respiration rate of $0.55 \pm 0.15$ mg C h$^{-1}$ kg$^{-1}$) ($P = 0.025$, SI, Table 3). Furthermore, the heterotrophic $CO_2$ respiration rate at elevation cluster of 1750-1850 m a.s.l. was $0.51 \pm 0.20$ mg C h$^{-1}$ kg$^{-1}$ soil, which was statistically similar to the respiration rate of $0.55 \pm 0.15$ mg C h$^{-1}$ kg$^{-1}$ soil at the highest elevation cluster (2700-3000 m a.s.l.). Similarly, heterotrophic respiration rate of $0.67 \pm 0.22$ mg C h$^{-1}$ kg$^{-1}$ soil and of $0.64 \pm 0.25$ mg C h$^{-1}$ kg$^{-1}$ soil were recorded for the lowest elevation cluster (1250-1300 m a.s.l.) and the mid-elevation cluster at 2500-2600 m
a.s.l., respectively (SI, Table 3). Consequently, heterotrophic $CO_2$ respiration rate from laboratory incubations showed a non-significant linear decrease at a rate of $0.004 \pm 0.008$ mg C h$^{-1}$ kg$^{-1}$ per 100 m of elevation increase ($R^2_m = 0.007$, $P = 0.677$, Table 2, Figure 3(a)).

In contrast, the specific heterotrophic $CO_2$ respiration rate (normalised per gram of SOC) revealed a significant linear trend
along the elevational transect. The highest $CO_2$ respiration rate of $17.2 \pm 5.3$ μg C h$^{-1}$ g$^{-1}$ SOC was detected at the lowest elevation cluster (1250-1300 m a.s.l.). This decreased to $10.8 \pm 4.8$ μg C h$^{-1}$ g$^{-1}$ SOC at 1750-1850 m a.s.l. and to $5.3 \pm 2.1$ μg C h$^{-1}$ g$^{-1}$ SOC at 2100-2200 m a.s.l. Furthermore, lower values of $3.7 \pm 1.9$ μg C h$^{-1}$ g$^{-1}$ SOC and $2.4 \pm 0.9$ μg C h$^{-1}$ g$^{-1}$ SOC were observed at the highest two elevation clusters of 2500-2600 m a.s.l. and 2700-3000 m a.s.l., respectively (SI, Table 3). Generally, along the elevation transect, the specific heterotrophic $CO_2$ respiration rate decreased linearly by $1.01 \pm$
$0.12$ μg C h$^{-1}$ g$^{-1}$ SOC per 100 m of elevation increase ($R^2_m = 0.68$, $P = 0.003$, Table 2, Figure 3(b)).
Further, following the decreasing trend in the specific heterotrophic $CO_2$ respiration rate along the elevational transect, respired $CO_2$ indeed was more depleted in $^{13}$C in warmer, lower elevations as compared to colder, higher elevations (Figure 3c). The $^{13}$C depletion factor of the respired $\delta^{13}CO_2$ relative to $\delta^{13}$C-SOC was $3.2 \pm 0.6$ ‰ at 1250-1300 m a.s.l., $2.8 \pm 0.9$ ‰ at 1750-1850 m a.s.l., $1.7 \pm 0.7$ ‰ at 2100-2200 m a.s.l., $1.0 \pm 1.3$ ‰ at 2500-2600 m a.s.l. and $-0.3 \pm 0.8$ ‰ at 2700-3000 m
a.s.l. (Figure 3(c), SI, Table 3). Along the elevational transect, the $^{13}$C depletion factor of the respired $CO_2$ showed a significant linear decrease by $0.23 \pm 0.04$ ‰ per 100 m of elevation increase ($R^2_m = 0.65$, $P = 0.011$, Table 2, Figure 3(c)).

On the other hand, along the elevational transect, AE ranged from $28.5 \pm 5.6$ kJ mol$^{-1}$ in the premontane elevation cluster (1250-1300 m a.s.l.) to $70.3 \pm 6.9$ kJ mol$^{-1}$ at 2500-2700 m a.s.l. and $69.9 \pm 3.0$ kJ mol$^{-1}$ in the highest elevation cluster
(2700-3000 m a.s.l.) (SI, Table 3). Generally, along the elevational transect, the AE showed a significant linear increase of $3.2 \pm 0.7$ kJ mol$^{-1}$ per 100 m of elevation increase ($R^2_m = 3.2$, $P < 001$). Similarly, along the elevational transect, $Q_{10}$ values ranged from $1.50 \pm 0.13$ in the lowest elevation (1250-1300 m a.s.l.) to $2.68 \pm 0.25$ in the highest elevation (2700-3000 m a.s.l), (SI, Table 3). Generally, along the elevational transect, the $Q_{10}$ showed a linear increase of $0.09 \pm 0.03$ per 100 m of elevation increase ($P = 0.012$, Table 2, Figure 3(d)).

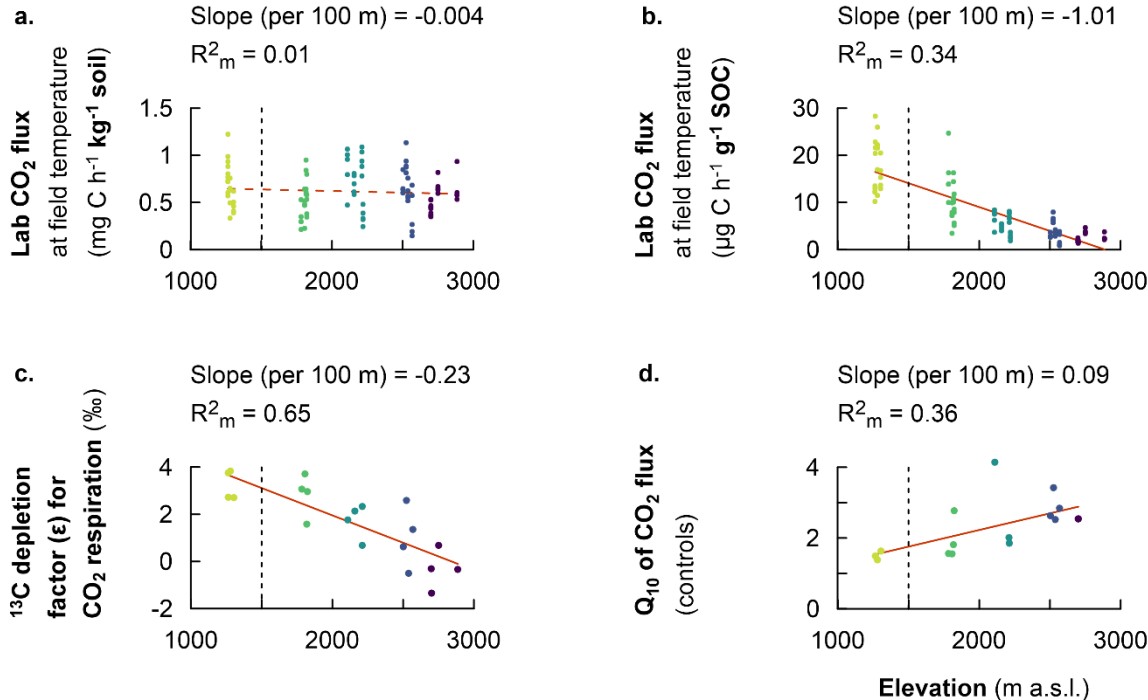


**Figure 3. Fixed effect estimates of elevation (per 100 m elevation increase) on response parameters: laboratory-based heterotrophic CO₂ respiration rates at corresponding mean *in situ* temperature (a), and the specific heterotrophic CO₂ respiration rates at corresponding mean *in situ* temperature (b), the ¹³C depletion factor for heterotrophic CO₂ respiration (c), and the temperature sensitivity of heterotrophic CO₂ respiration rates (Q₁₀) (d). The slope of the linear mixed effect model estimates per 100 m of elevation increase is indicated (red solid line for a significant effect and red dashed line for no significant effect), as well as the marginal R² (R²ₘ), representing the fraction of the response variable explained by elevation. Plots from montane forest clusters (from 1750-3000 m a.s.l.) were compared with a nearby premontane forest (separated by vertical dashed line) at an elevation of 1250-1300 m a.s.l.**






### 3.3 Effect of soil warming on $CO_2$ respiration, AE, $Q_{10}$, SOC and $\delta^{13}C$

After about 2 years of *in situ* soil warming, heterotrophic $CO_2$ respiration rates, AE and $Q_{10}$ were assessed in a laboratory incubation experiment for control and warmed soil. Additionally, the SOC content and its $\delta^{13}C$ for control and warmed soil were analysed. The results revealed that at each elevation cluster, there was no significant difference in the studied parameter between control and warmed soil (SI, Table 3). However, along the entire elevational transect, both the non-specific and specific heterotrophic $CO_2$ respiration rates for controlled soil were relatively higher than those of warmed soil (Figure 4a

and b respectively, SI, Table 3). Similarly, along the elevational transect, both the AE and $Q_{10}$ coefficients for control were relatively higher than those of warmed soil (Figure 4c and d respectively). Additionally, after two years of *in situ* soil warming, the SOC contents of warmed soil were relatively lower than those of control along the elevational transect (Figure 4(e)). Finally, the $\delta^{13}C$ composition of the SOC showed that warmed soil became relatively more enriched in $^{13}C$ compared to control soil (Figure 4(f)).





**Figure 4. Comparison of warming response of soil organic carbon along the Rwenzori elevational transect; heterotrophic CO₂ respiration rates of control and warmed soils (measured at the translocated *in situ* temperature) (a), idem for the specific CO₂ respiration rates (b), activation energy for control and warmed soil (c), sensitivity of CO₂ respiration rates to temperature (Q₁₀) for control and warmed (d), soil organic carbon content for control and warmed soil (e), and δ¹³C composition of soil organic carbon for control and warmed soil (f).**





### 3.4 *In situ* CO₂ respiration under current conditions at start and mid rainy season

The *in situ* total (heterotrophic and autotrophic) $CO_2$ respiration rates along the elevational transect at the start of the rainy season were only significantly different when comparing the upper montane cluster (2700-3000 m a.s.l.) with the rest of the

elevation clusters (1250-2200 m a.s.l.) ($P = 0.027$, SI, Table 3). Total $CO_2$ respiration rate ranged from $95.1 \pm 34.6$ mg C m$^{-2}$ h$^{-1}$ in the lower montane elevation at 2100-2200 m a.s.l. to $59.3 \pm 16.7$ mg C m$^{-2}$ h$^{-1}$ at 2700-3000 m a.s.l.. The total $CO_2$ respiration rate in the lower three elevations (1250-2200 m a.s.l.) were similar, ranging between $79.2 \pm 17.3$ mg C m$^{-2}$ h$^{-1}$ to $95.1 \pm 34.6$ mg C m$^{-2}$ h$^{-1}$ (SI, Table 3). Generally, along the elevational transect, there was no significant linear trend in the total $CO_2$ respiration rate, though this tended to decrease at a rate of $1.04 \pm 1.49$ mg C m$^{-2}$ h$^{-1}$ per 100 m of elevation increase

at the start of rainy season ($R^2_m = 0.04$, $P = 0.558$, Table 2, SI, Figure 3).

On the other hand, in the mid rainy season, there was a much stronger variability in the *in situ* total $CO_2$ respiration rates among the elevation clusters ($P < 0.001$, SI, Table 3). The total $CO_2$ respiration rate was highest (i.e. $113.2 \pm 35.7$ mg C m$^{-2}$ h$^{-1}$) in the premontane elevation cluster at 1250-1300 m a.s.l.. A statistically similar respiration rate of $112.8 \pm 20.3$ mg C m$^{-2}$

h$^{-1}$ was found in the montane foothill at 1750-1850 m a.s.l.. There was a significantly lower total $CO_2$ respiration rate (i.e. $89.0 \pm 22.3$ mg C m$^{-2}$ h$^{-1}$) at 2100-2200 m a.s.l., which further decreased to $67.7 \pm 9.6$ mg C m$^{-2}$ h$^{-1}$ at 2700-3000 m a.s.l. (SI, Table 3). Overall, along the elevational transect, there was also no significant linear trend in the total $CO_2$ respiration rate which tended to decrease at a rate of $3.25 \pm 0.89$ mg C m$^{-2}$ h$^{-1}$ per 100 m of elevation increase ($R^2_m = 0.33$, $P = 0.067$, Table 2, SI, Figure 3).


Comparison of seasonal *in situ* total $CO_2$ respiration rates generally revealed that higher soil moisture contents were associated with higher $CO_2$ respiration rates. Specifically, at 1250-1300 m a.s.l., total $CO_2$ respiration rate at the start of the rainy season ($80.1 \pm 15.8$ mg C m$^{-2}$ h$^{-1}$) increased significantly to $113.2 \pm 9.6$ mg C m$^{-2}$ h$^{-1}$ in the mid rainy season ($P < 0.001$, SI, Table 3). Similarly, the total $CO_2$ respiration rate at 1750-1850 m a.s.l. increased significantly from $79.2 \pm 17.3$

mg C m$^{-2}$ h$^{-1}$ at the start of the rainy season to $112.8 \pm 20.3$ mg C m$^{-2}$ h$^{-1}$ in the mid rainy season ($P < 0.001$). There was no seasonal variation in the total $CO_2$ respiration rate at 2100-2200 m a.s.l., where the total $CO_2$ respiration rate was between $95.1 \pm 34.6$ mg C m$^{-2}$ h$^{-1}$ and $89.0 \pm 22.3$ mg C m$^{-2}$ h$^{-1}$ in the start and the mid rainy season, respectively. At 2700-3000 m a.s.l., the total $CO_2$ respiration rate increased significantly from $59.3 \pm 16.7$ mg C m$^{-2}$ h$^{-1}$ in the start of rainy season to $67.7 \pm 9.6$ mg C m$^{-2}$ h$^{-1}$ in the mid rainy season ($P = 0.035$, SI, Table 3).




**Table 2. Fixed effect estimates of elevation (per 100 m elevation increase) on heterotrophic $CO_2$ respiration rates under laboratory incubations at corresponding mean *in situ* temperature, laboratory-based heterotrophic specific $CO_2$ respiration (normalised per unit soil organic carbon (SOC)), $^{13}C$ depletion factor for $CO_2$ respiration ($\varepsilon$), and temperature sensitivity of $CO_2$ respiration rate ($Q_{10}$) for control and warmed soil, and *in situ* total $CO_2$ respiration rates at the start and in the mid rainy season along the**

**elevational transect. The associated standard error (SE), *P*-value, marginal coefficient of determination ($R^2_m$) and conditional coefficient of determination ($R^2_c$) are indicated.**

| Parameters | Effect estimate | SE | *P*-value | $R^2_m$ | $R^2_c$ |
|---|---|---|---|---|---|
| Laboratory-based $CO_2$ respiration ($\mu g$ $CO_2$-C $h^{-1}$ $kg^{-1}$ soil) | -0.00 | 0.01 | 0.680 | 0.01 | 0.12 |
| **Laboratory-based specific $CO_2$ respiration (mg $CO_2$-C $h^{-1}$ $g^{-1}$ SOC)** | **-1.01** | **0.12** | **0.003\*** | **0.68** | **0.72** |
| **$^{13}C$ depletion factor for $CO_2$ respiration ($\varepsilon$) (‰)** | **-0.23** | **0.04** | **0.011\*** | **0.65** | **0.66** |
| **$Q_{10}$ control** | **0.09** | **0.03** | **0.012\*** | **0.36** | **0.36** |
| **$Q_{10}$ warmed** | **0.09** | **0.03** | **0.013\*** | **0.37** | **0.37** |
| $CO_2$ respiration at start of rainy season (mg $CO_2$-C $h^{-1}$ $m^{-2}$) | -1.04 | 1.49 | 0.558 | 0.04 | 0.35 |
| $CO_2$ respiration in mid rainy season (mg $CO_2$-C $h^{-1}$ $m^{-2}$) | -3.25 | 0.89 | 0.067 | 0.33 | 0.40 |

A statistically significant elevational linear trend is bolded and *P*-value marked with asterisk symbol "\*", $R^2_m$ is the proportion of the variance in the response variable explained by the fixed effect (elevation), $R^2_c$ is the proportion of the variance in the response variable explained by the fixed effect plus random location effects.




## 4 Discussion

### 4.1 Elevational trend in $CO_2$ respiration, $Q_{10}$ and $^{13}C$ depletion factor (Epsilon)

The specific heterotrophic $CO_2$ respiration decreased along elevation in part, due to negative effect of low temperature on microbial activity (Zimmermann et al., 2009). In support of the temperature effect on $CO_2$ respiration (Figure 3(b)), the $^{13}C$

fractionation factor for heterotrophic $CO_2$ respiration was also temperature-dependent, such that the respired $CO_2$ at the warm, lower elevations showed a higher fractionation, and subsequently became relatively more depleted in $^{13}C$ than at the cold, higher elevations (Figure 3(c)). This also indicates that the higher the specific heterotrophic respiration rates, the higher the discrimination against the heavier $^{13}C$ isotope (Andrews et al., 2000; Natelhoffer and Fry, 1988). These results imply that at higher elevations, even though SOC contents were high, microbial SOC decomposition was limited by low temperatures

(Zimmermann et al., 2009). In addition, low $CO_2$ respiration at high elevations is explained by the inhibitory effect of the low soil pH on microbial respiration (Figure 3(c), SI, Table 1) (Rousk et al., 2009), by impairing microbial activity (Walse et al., 1998). Further, the low pH also facilitates the stabilisation of organic matter through complexation reactions with iron and aluminum ions, which become soluble at a low pH (Lützow et al., 2006).

Additionally, the observed linear increases in $Q_{10}$ and AE with elevation (Table 2, Figure 3(d)), indicate an increasing trend in soil organic matter recalcitrance resulting in lower $CO_2$ respiration at high elevations (Davidson and Janssens, 2006). This is corroborated by the decreasing $^{13}C$ fractionation (Table 2, Figure 3(c)), as well as lower specific heterotrophic $CO_2$ respiration in higher elevations at a uniform temperature and 60 % WFPS (r= -0.53, $P$ = 0.02, SI, Table 2, SI, Figure 5). The linearly increasing elevational trend in AE and $Q_{10}$ is consistent with the intrinsic principles of microbial respiration kinetics

associated with the Arrhenius equation (Craine et al., 2010; Davidson and Janssens, 2006; Schipper et al., 2014). Accordingly, the sensitivity of microbial decomposition to an increment in temperature should increase with increasing AE and recalcitrance of the substrate (Craine et al., 2010; Luo et al., 2001; Davidson and Janssens, 2006). As such, more complex (i.e. recalcitrant) C compounds, for which the microbial decomposition requires more AE, facilitates a lower specific heterotrophic $CO_2$ respiration (SI, Figure 5) and a higher $Q_{10}$ (i.e. their decomposition is enhanced to a higher extent

at a certain increase in temperature) than more simple (i.e. labile) C substrates (Craine et al., 2010; Davidson and Janssens, 2006; Nottingham et al., 2019).

### 4.2 Effect of *in situ* warming

Generally, after two years of *in situ* warming, $\delta^{13}C$ of SOC revealed a relative enrichment in $^{13}C$ isotope in warmed soil as compared to the control (Figure 4(f)). This is due to enhanced mineralisation rates in the warmed soil. Higher mineralisation

causes stronger $^{13}C$ fractionation (Amundson et al., 2003), due to microbial discrimination against $^{13}C$ during C transformation processes (Andrews et al., 2000; Natelhoffer and Fry, 1988). These results imply that warming indeed



increased mineralisation of SOC in warmed relative to control soil, owing to the increase in microbial activity at higher temperatures in correspondence with the Arrhenius equation (Craine et al., 2010; Mohan, 2019; Nottingham et al., 2020). Indeed along the elevational transect, SOC was relatively lower in warmed as compared to control (Figure 4(e)).

Subsequently, the $CO_2$ respiration for warmed soil were relatively lower than in the controls due to the apparent depletion of respiration substrate during the two years of warming. The evidence of accelerated mineralisation of SOC upon warming supports the results of several soil warming studies that reported an increment in $CO_2$ respiration upon warming (Eliasson et al., 2005; Luo et al., 2001; Melillo et al., 2002; Rustad et al., 2001). The increased decomposition following a temperature increase has the ability to inherently change the content and quality of the SOC, if organic inputs cannot replenish SOC loss

at the same rate (Craine et al., 2010; Davidson and Janssens, 2006; Luo et al., 2001). Nonetheless, the lack of significant difference in $CO_2$ respiration rates for warmed and control soil at each elevation cluster after two years of warming may also indicate the fact that labile SOC fractions were not depleted after two years at a higher temperature (Crowther et al., 2016; Davidson and Janssens, 2006; Melillo et al., 2017).

On the other hand, along the elevational transect, the AE and $Q_{10}$ values of warmed soils were relatively lower than in the control soils (Figure 4(c) and (d) respectively). This trend is expected as the Arrhenius equation predicts that the $Q_{10}$ of a certain reaction intrinsically decreases with increasing temperature (Craine et al., 2010; Davidson and Janssens, 2006). As such, SOC substrates would become overall less recalcitrant after two years at an elevated temperature. Indeed, while warming increases the mineralisation rate of the most labile C, it also stimulates the decomposition of recalcitrant C fractions

to a relative higher degree than labile fractions because of the higher $Q_{10}$ of the former (Craine et al., 2010; Davidson and Janssens, 2006). Subsequently, the AE and $Q_{10}$ of warmed soils were relatively lower than those of the controls (Davidson and Janssens, 2006; Mohan, 2019; Nottingham et al., 2020). Both the decrease in SOC contents and the relatively enhanced activation of recalcitrant organic matter upon warming undermine the climate mitigation function of the soil (Davidson and Janssens, 2006; Nottingham et al., 2020; Walker et al., 2018).

### 505 4.3 Present-day $CO_2$ respiration along the Rwenzori elevational transect

Along the Rwenzori Mountains elevational transect, we observed that the *in situ* total $CO_2$ respiration rates were significantly lower at the highest elevation cluster of the montane forest in both the start and the mid rainy season as compared to premontane forest. On the other hand, by isolating the effect of moisture in a laboratory incubation (at uniform 60 % WFPS and corresponding *in situ* measured temperatures per elevation cluster), we found similar heterotrophic $CO_2$

respiration along the transect (SI, Table 3, Figure 3(a)), and ultimately we confirmed a significant linearly decreasing trend for the specific heterotrophic $CO_2$ respiration (Figure 3(b)). This indicates that, under *in situ* condition, low temperatures and low soil moisture contents limited microbial $CO_2$ respiration at high elevations. Indeed low temperatures at high elevations can have a negative effect on decomposer activities (Finzi et al., 2006; Luo et al., 2006). Meanwhile, an adequate soil





moisture content can boost microbial respiration by facilitating both the diffusion of soluble substrates and the transport of
oxygen (Liu et al., 2006). Generally, the optimal soil moisture content for microbial respiration is reported to be
approximately 60 % WFPS (Aon et al., 2001; Doetterl et al., 2015). Subsequently, in the mid rainy season (when the soil
WFPS increased) (SI, Table 3), in correspondence, we detected an increase in the total *in situ* $CO_2$ respiration rate in all
elevation clusters, but at higher elevations between 2100-3000 m a.s.l., the increase in $CO_2$ respiration rates were not
significant owing to only a small increase in WFPS in the mid rainy season (SI, Table 3).

The microbial community structure indicated that though the variation of the microbial community (in terms of bacteria and
fungi), as a function of elevation is rather limited (SI, Figure 4), there is some shift. In particular, higher elevations are
relatively more dominated by gram-positive bacteria (Figure 4, SI, Table 2) and by fungi relative to bacteria (SI, Figure 4),
which is also further indictor for higher organic matter recalcitrance (Fanin et al., 2019; Lipson et al., 2002). Based on the
principal component analysis and linear mixed effect model analysis of microbial structure (Figure 4, SI, Figure 4), the ratio
of total bacteria to fungi slightly decreased with increasing elevation, where it is also known that fungi are often relatively
more dominant in soils characterised by more complex organic materials (Lipson et al., 2002), as fungi are more specialised
in the breakdown of recalcitrant organic matter (Boer et al., 2005; Coleman et al., 2017). Altogether, we showed that
microbial $CO_2$ respiration along the elevational transect was limited by an increasing carbon recalcitrance, decreasing soil
temperature, moisture content and pH.

## 5 Conclusion

Our results indicated that global warming can lead to enhanced losses of SOC in montane forests due to their increasing
temperature sensitivity and SOC content with elevation. Therefore, the high concentrations of SOC at higher elevations of
montane forests are particularly at stake, since the climate warming exactly undermines the mitigating effect of low
microbial decomposition under; (i) low temperature (ii) high organic matter recalcitrance (due its higher temperature
sensitivity). Further, we showed that along the elevational transect, *in situ* warming indeed led to increased SOC
mineralisation and the associated isotopic fractionation resulted in a relative enrichment of [13]C isotope in warmed soils as
compared to the controls. This is due to stronger discrimination against the [13]C isotope owing to faster SOC transformation
processes. Ultimately, along the elevational transect, after two years of warming, the SOC in the warmed soils were
relatively lower than in the controls, indicating a depleting trend in SOC owing to the increment in mineralisation during the
two years at higher temperature.



**Acknowledgement**

This work was funded by VLIR-OUS and Mountains of the Moon University under the partnership program of Inter-
545 University Cooperation (IUC), grant number UG2019IUC027A103. The authors thank Uganda Wildlife Authority for
granting permission to conduct this study in two protected National Parks under permit number UWA/COD/96/05. We also
thank staff of the Uganda Wildlife Authority at Rwenzori Mountains and Kibale Forest National Parks for their support.
Further, we are grateful to the Research Assistant, Mr. Martin Tuisenge for his tireless efforts and endurance during the
demanding field campaigns in the tremendously physically challenging Rwenzori Mountains.

**Disclosure statement**

The authors declare that they have no conflict of interest





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
