# Peer review of "Temperature sensitivity of soil organic carbon respiration along a forested elevational gradient on the Rwenzori Mountains, Uganda"

_Biogeosciences, 2022_

## Author Comment (AC1)

**Comment on bg-2022-37**

**Anonymous Referee #1**

**Referee comment on "Temperature sensitivity of soil organic carbon respiration along the Rwenzori montane forests elevational transect in Uganda" by Joseph Okello et al., Biogeosciences Discuss., https://doi.org/10.5194/bg-2022-37-RC1, 2022**

**The manuscript by Okello et al. presents a potentially interesting dataset examining the sensitivity of soil organic carbon stocks to projected temperature changes. The work fits well in the scope of BG, and I much appreciate the important work done – but the presentation of the data and the interpretation does require a substantial amount of clarification and improvement before being reconsidered. Detailed comments below.**

*Dear reviewer, thank you very much for appreciating our work and equally for the meticulous review and insightful suggestions to further improve the manuscript. In the following specific sections, we pay attention to address the concerns raised.*

**Main comments & suggestions**
**-Terminology:**

**\*throughout the manuscript, the terminology related to stable isotopes is really not OK. For example, the authors refer to "the $^{13}$C depletion factor" or "isotopic depletion factor" (L167) – that is not an accepted term in the literature, what you are referring to is termed fractionation (epsilon).**

*Thank you for this remark. Indeed, as you correctly identified, our study focused on isotopic fractionation (epsilon), i.e. the change in the stable isotope composition of C during the transformation of SOC to emitted $CO_2$, as a result of discrimination against $^{13}$C during SOC transformation that involved physical and chemical processes.*

*We indeed also realised that, strictly speaking, it was incorrect to refer to epsilon as a factor, because in contrast to "alpha" (which we no longer mention in the new version of the manuscript), epsilon is a difference.*

*Further, we agree to rearrange Equation 2 in the new version of the manuscript as follows:*

"
$$\varepsilon = \left( \left( \frac{\text{R-CO}_2}{\text{R-SOC}} \right) - 1 \right) * 1000 \qquad (2)$$

Where:

R-CO$_2$ is the ratio of $^{13}$C to $^{12}$C in the emitted CO$_2$

R-SOC is the ratio of $^{13}$C to $^{12}$C in soil organic carbon (substrate)."

**\*other examples: L 321 "$^{13}$C and content of soil organic carbon relatively increased"**

*We agree to revise this in the new version of the manuscript as follows: "Further, in warmed soils, the $\delta^{13}$C values and soil organic carbon content increased and decreased, respectively."*

**\*Keeling mass balance approach (line 155-162): this is just a mass balance approach, a Keeling plot is something quite different; equation 1 does not appear in the Keeling (1958) paper you refer to.**

*Thank you for this observation. Indeed, we agree that this is simply a mass balance approach and we shall add citations of this approach. While indeed Keeling (1958) used this approach in the Keeling plot method over several points of measurements, we used the mass balance approach for only two CO$_2$ measurement points. As such we just calculated the mass balance using equation 1 (line 157). We agree to rephrase this statement in the new version of the manuscript as follows:*

*"To determine the $\delta^{13}$C values of the respired CO$_2$, we used a mass balance approach (Phillips and Gregg, 2001).*

$$\delta^{13}\text{C-CO}_2 \approx \frac{[\text{CO}_2]_{final} * \delta^{13}\text{C-CO}_{2\,final} - \text{CO}_2]_{initial} * \delta^{13}\text{C-CO}_{2\,initial}}{[\text{CO}_2]_{final} - [\text{CO}_2]_{initial}} \qquad (1)"$$

**\*L 146, 190, 289: "$\delta^{13}$C isotopic composition": again, this is not appropriately phrased. Use either "$\delta^{13}$C  values" or "the C stable isotope composition" but not combinations of the two.**

*We agree to rephrase throughout the manuscripts and use the term "$\delta^{13}C$ values". For instance for those specific lines:*

*L146. "Immediately, one gas sample was taken using a 45 mL syringe. The ambient $CO_2$ concentration and its $\delta^{13}C$ value at "open condition" was analysed using Cavity Ring-Down Spectrometer, (G2113-I, CRDS $CO_2$ analyser, Picarro, United States)."*
*(The "open condition" is the initial sampling time ($T_0$))*

*L190. "The top 10 cm of the soil cores was collected (i.e. the soil layer with the highest C content and most active in C cycling), homogenised, air-dried, and sieved (2 mm mesh size) for additional laboratory incubation experiments, in order to assess the effect of two years of in situ warming on: (i) $CO_2$ respiration rates; (ii) the AE and $Q_{10}$ coefficient; and (iii) SOC content and $\delta^{13}C$ values."*

*L289. "Further, to check for changes in $CO_2$ emission rates, AE, $Q_{10}$, SOC content and $\delta^{13}C$ values between control and in situ warmed soil at each elevation cluster, we used a Wilcoxon test."*

**\*enrichment in $^{13}C$ isotope (L478): enrichment in $^{13}C$**

*We agree to rephrase as suggested.*

**\*L373: $\delta^{13}CO_2$ --> $\delta^{13}C$ -$CO_2$ or $\delta^{13}C$ of $CO_2$**

*Thank you for the observation, we agree to rephrase this to "$\delta^{13}C$-$CO_2$" in the new version of the manuscript.*

**-PLFA data: there is a short section in the Methods outlining the extraction and derivatization of PLFA, and then basically nothing. No info on how PLFA were identified and quantified, no information on how the resulting data were treated – assigned to microbial groups etc. The data are presented later on as concentrations of PLFA representing gram-positive, gram-negative bacteria, fungi, etc. but no information or references are given; no mention of this in the Methods, and very little real discussion of**

**these data. Either add all this info, or remove them if the data don't contribute much to the story.**

*Thank you for this insight. We agree to add some procedures and citations of the method used from line 132. i.e. " Eventually, the phospholipid fatty acids were converted to methyl esters, which were subsequently analysed using gas chromatography (GC, Trace GC, Thermo Scientific, Bremen, Germany)" following the methods described by Denef et al. (2007) and Huygens et al. (2011).*

*Additionally we agree to add the following: "We determined the ratios of the peak area of each individual PLFA to that of C16:0, a universal PLFA occurring in the membranes of all organisms. PLFA ratios less than 0.02 were excluded from the data set (Drijber et al., 2000). PLFA was assigned to microbial fungal group following (Zelles, 1997) and (Chung et al., 2007). While PLFA assignment to bacterial group and to gram-positive and gram negative bacteria followed the procedure described by Kroppenstedt (1985) and Frostegård and Bååth (1996)"*

**-precipitation: L101:7000 mm should be 700mm probably or something in that order of magnitude at least. The study sites cover a wide range of precipitation, and there is also a clear difference between Kibale and Rwenzori (as in: precipitation is higher in Kibale than at the lowest elevation along the Rwenzori transect)- however, the effect of precipitation on the data from the translocation experiments is not discussed at all; this should be worth some discussion.**

*We agree that there must have been an error in precipitation data at elevation of 1760 m. The available rainfall data from the Uganda Wildlife Authority (2012-2016) indicated unusually high rainfall at two of the six elevations where data were collected. When picking the ranges of rainfall, we selected one of these high values. We have now used the ranges to only include rainfall data for the realistic four data points (all with rainfall amounts between 1500-1800 mm per annum). As such, rainfall ranged from 1570 mm at 1760 m to 1806 mm at 4230. Therefore we propose to use these rainfall amounts in the new version of the manuscript.*

*Further, while we used this data to give an indication of rainfall in the Rwenzori Mountains, the data are not specific to our study plots (only one weather station is within the studied site elevation). Therefore we cannot conclusively tell the rainfall trend in the elevational gradient.*

*Due to this uncertainty, in our study plots we focused on directly monitoring the soil moisture content which indeed directly affects microbial activities. We discussed the effect of soil moisture elaborately. For this reason, we did monitor $CO_2$ efflux under in situ conditions in two key periods (start of rain and mid rainy season). Indeed, the results indicated an increase in $CO_2$ efflux following increase in soil moisture in the mid rainy season. Because of this we noted that soil moisture has an effect on $CO_2$ efflux (see section 4.3 line 506-519). Additionally, under laboratory condition when moisture content was standardized, we noted that $CO_2$ emission did not differ along the elevational gradient, but decreased linearly when standardised per amount of SOC (i.e., effect of temperature is isolated, line 508-519).*

**-L111-116: No mention is made on whether samples were acidified to remove potential carbonates (or carbonates precipitation from the soil solution during soil drying). C/N ratios are reported but you need to specify whether these are weight/weight or molar ratios. For a proper interpretation of data, specify the reproducibility of your measurements (e.g. for $\delta^{13}C$ ) and mention which standards were used. In Table 1, specify then if these concentrations refer to organic or total C. If total C, then you may want to add a note of caution in the interpretation of differences between $\delta^{13}C$ of soil C and $CO_2$ produced.**

*Thank you for these remarks. Practically at the prevailing soil pH-KCl (5.4-3.3, see table 1), the soil is quite acidic, and the presence of carbonates is negligible. Secondly, in the wet tropics (where there is precipitation surplus), carbonates are commonly dissolved and leached down to deeper soil layers, yet we only sample the top 30 cm of soil. Given these reasons, the total carbon measured was taken as the soil organic carbon. Therefore, agree to explicitly mention this in the revised version of the manuscript in the methodology section from line 116.*

**-section 2.3: provide a description of how temperature was controlled during these incubations.**

*We agree with this comment, we should have mentioned that we used an incubator earlier in section 2.3 (it is in line 144). In the new version of the manuscript, we agree to mention it at the beginning of in line 140 as follows:*

*"The samples were then pre-incubated in the incubator for 14 days (at the respective mean annual temperature per elevation cluster, i.e. 20, 17, 15, 13 and 12 °C for elevation clusters of 1250-1300, 1750-1850, 2100-2200, 2500-2600 and 2700-3000 m a.s.l., respectively)."*

**-section 2.3: mention how you coped with removing a 45 mL gas sample from your incubation jars: was this volume replaced with air while taking these samples, if not how was the pressure difference accounted for in your measurements?**

*We agree that this is needed. We wish to clarify that in the in situ $CO_2$ measurements, the gas chamber was equipped with a vent tube to minimize pressure differences (mentioned in line 220-221). Additionally, for in situ $CO_2$ measurements, only 15 mL gas sample was taken at a time (see line 227-228),( yet the total volume of the chamber was about 4 L). While for laboratory incubation, headspace gas sampling for the determination of soil $CO_2$ fluxes was done twice, directly at the beginning before fully closing the jar and at the end of the incubation period. For gas sampling a gas tight syringe (45 mL) was used, which was immediately gas-tightly closed after taking the gas sample. Gas samples were analyzed immediately using Cavity Ring-Down Spectrometer, (G2113-I, CRDS $CO_2$ analyser, Picarro, United States). Thus, during incubation the pressure in the jar was not affected. Moreover, immediately after the second sampling, the jars were opened and only covered with parafilm to equilibrate until the following day (see line 151-153).*

**-L208: why refer to the "slope of the $CO_2$ concentration in function of time"? If I understand well, you simply have measurement at the start and end of the incubation ?**

*We appreciate this suggestion. Indeed, this we agree to rephrase as follows: "Eventually, the $CO_2$ emission rate was determined as change in headspace $CO_2$ concentrations ($t_1$-$t_0$) divided by the incubation time (24 hours) for laboratory experiments. Still for the in situ measurements we calculated a slope representing increase of $CO_2$ concentrations (N = 5) over chamber closure time (90 minutes)".*

**-L199 and further: any reason to go for 50 mL jars here instead of 1 L jars (as in section 2.3) ? For the $\delta^{13}C$ measurements, it's important to convince the reader that the data you collected from these experiments are valid: you are in a closed system, where you sample gaseous CO2 for $\delta^{13}C$ analysis, but you also have an aqueous phase. $CO_2$ will equilibrate**

**between the two, and there is a (small) degree of isotope fractionation involved. The smaller the headspace volume compared to the volume of soil (and thus water), the higher the possible bias in resulting δ¹³C -CO₂ data if not accounted for. It might be negligible in your setup, but you need to provide arguments to show this.**

*Thank you for this comment. We are aware of the isotopic fractionation between headspace and liquid phase, though this section you refer to is only about $CO_2$ headspace concentration measurements, but not isotopes. Further, as clarified in this experiment, firstly, we ensured that the headspace volume and moisture content was constant for all the samples. Secondly, in preliminary experiments we ensured that headspace $CO_2$ concentrations increased linearly over a time period of 24 hours.*

**-Section 2.5: while I am aware that much of the literature refers to soil CO₂ flux measurements using closed chambers as "soil respiration", one should avoid keeping using this terminology in use; what is measured is not total in situ soil respiration but the diffusive flux of CO₂ from the soil. This diffusive flux is governed by the gradient of CO₂ concentrations / partial pressures and is thus influenced by e.g. porosity, water content etc. Part of the CO₂ produced by soil respiration is lost via percolation and groundwater losses.**

*We concur with this statement. In the new version of the manuscript, we agree to explicitly mention in line 211 that we measured the in situ diffusive flux of $CO_2$ from the soil ($CO_2$ emission) and used it as a proxy for $CO_2$ respiration (hereafter referred to as "$CO_2$ emission").*

**-Chamber deployment time (L224): 90 minutes seems excessively long for chamber closures, especially given the low chamber surface area. Pavelka et al. (2018, doi:10.1515/intag-2017 0045) and other recommend much lower chamber closure times, in the order of 5 minutes. Were the chambers equipped with a fan to ensure proper mixing within the chambers ? If the CO₂ increase was not linear, this has implications for your fluxes as well as δ¹³C data interpretation.**

*Thank you for this comment. We are aware of the effect of increasing gas concentrations in the headspace on $CO_2$ diffusive fluxes out of the soil. While shorter chamber closure times are indeed preferable, this could not be realised under field conditions. At the time of the field*

*campaigns, we did not have the possibility to do continuous measurements in situ (as the instrument broke down), so this was the best alternative. Further, we purposely took headspace air samples every 30 minutes interval, so that it was possible to check linearity with time and indeed the CO$_2$ concentration was always linear over the 90 minutes intervals (see line 204-206). Secondly, the gas chambers were fitted with a vent tube to minimise any changes in pressure. As such, the conditions for a non-steady state closed chamber flux measurements were still respected. We agree to explicitly mention this statement in the new version of the manuscript from line 229.*

**Not clear, by the way, if all $\delta^{13}$C -CO$_2$ measurements were made using a Picarro G2113, this is only mentioned for the t0 samples in section 2.3. If other CO$_2$ samples were measured using other methods, add the necessary info (equipment, standards, reproducibility) to your Methods section.**

*Thank you for this comment. For clarification, in the laboratory incubation in section 2.3, we used Picarro G2213 for both T0 and T24 as mentioned (line 146-147). While for section 2.4 (about the soil mesocosm translocation experiment), we used a gas chromatograph (Finnigan Trace GC Ultra, Thermo Electron Corporation, Milan, Italy) fitted with a thermal conductivity detector (mentioned in line 205-207). The gas chromatography measurements were only for CO$_2$ concentration and not for $^{13}$C.*

**-L237: linear regression: if you have 2 data points only, then avoid referring to this as 'fitting a linear regression to the concentrations over time'.**

*Thank you for this suggestion. We concur and adapted as above in the comment about L208.*

**-L402-403: 'the SOC contents of warmed soil were relatively lower than those of control (soils) along the elevational transect': While Figure 4e may indeed suggest this, this does not appear to be the case for the lower elevation sites + even for the higher elevations the large error bars do not suggest that this difference is significant. Quantifying small changes in SOC stocks is challenging – if the difference is not statistically significant, then avoid phrasing in the way it is currently done. If you do feel confident that these are robust differences, you need to provide statistical justification + provide an estimated analytical**

**error on your bulk density and %C (or OC, see elsewhere) data. The same comes back on L 484 where you claim that SOC was relatively lower in warmed as compared to control [samples] – if these differences are not significant then such statements should be rephrased; this is what others will pick out as conclusions in subsequent work.**

*Thank you for this insight. Indeed we agree with this comment. For this reason, we first mentioned the following: "Results revealed that at each elevation cluster, there was no significant difference in the studied parameter between control and warmed soil (SI, Table 3)" (line 395-398)."*

*The high spatial and temporal variability in measurements of soil diffusive $CO_2$ fluxes often preclude powerful statistical tests of small treatment effects (Davidson and Janssens, 2006), as is usually the case in warming experiments. This effect is even more critical in montane ecosystems with different slope positions and aspects. Therefore, a parallel linear regression model fit along the entire elevation can help to reveal some "trend" between warmed and control treatments for soils taken at different elevations. To clarify further, these parallel linear model only reveals some trend but doesn't test for significant differences. The regression slopes indeed revealed a trend between control and warmed samples when fitted side by side.*

*Finally, we have added the statistics for SOC and $\delta^{13}C$ and also the slopes and intercepts of the linear mixed effect model for SOC and $\delta^{13}C$ for both control and warmed treatments in SI, Table 3.*

**-L403-404: the data presented in Figure 4f show a surprisingly large difference in $\delta^{13}C$ values, albeit with relatively high standard deviations. It would be good to provide statistics for this: for which elevations are these differences significant or not ? Again, I assume that for the high elevation sites, they are not statistically different – which should not be a surprise, as you have very organic soils here for which you would need to have a very high turnover rate to see any differences in $\delta^{13}C$ of the SOC pool after 2 years. You could likely do some back-of-the envelope calculations here.**

*Thank you for this insight. We agree with this comment. As mentioned earlier, L402-403 and L484, there were no significant differences between control and warmed samples for a given elevation, while using linear mixed effect model regression analysis across entire elevational gradient allowed to show a trend between warmed and control treatments. As such we agree to*

*clarify in the new version that "the remaining SOC tended to be more enriched in $^{13}C$ in warmed than in control, and the SOC tended to be lower in warmed than in control treatment".*

*We agree to also add the statistics for SOC and $\delta^{13}C$ in SI, Table 3 along with the slopes and intercepts of the linear mixed effect model regression fit in the new version of the manuscript.*

**-Discussion, section 4.1. The discussion on differences in $\delta^{13}C$ between $CO_2$ and soil (organic) carbon needs to be reconsidered. The current discussion assumes that there should be a relationship between rates of mineralization and isotope fractionation during respiration. On line 480, you refer to Amundson et al. (2003) to back up this idea – but this is a paper that only discusses nitrogen stable isotope ratios in soils. As far as I'm aware, there is no sound evidence in the literature that the degree of isotope fractionation (if any) during respiration would be related to either respiration rates, or temperature – as you hint at in the first paragraph of section 4.1. You also refer to Andrews et al. (2000) and Natelhoffer & Fry (1988) here, but these do not really back up such statements: (i) Natelhoffer & Fry merely demonstrate that the SOC pool is typically enriched in 13C as mineralization progresses, without unambiguously demonstrating via which mechanisms (selective mineralization or degradation, Suess effect etc – for an updated discussion see e.g. Ehleringer et al. 2000 Ecological Applications 10: 412-422 and subsequent literature); and (ii) Andrews et al. (2000) should be interpreted carefully here. Granted, they observed similar patterns for soils from FACE experiments and control soils, but note that they do not invoke kinetics in offering an explanation to their data: "The increase in respiration rate across the entire temperature range and the enrichment in $^{13}C$ only at 4°C rule out a strictly kinetic explanation for the observed carbon isotope fractionation. In addition, there is no theory that suggests a very different ratio of reaction rates of $^{13}C$ compared to $^{12}C$ in slow versus fast reactions (Agren et al., 1996). We suggest that the shift in carbon isotopic ratios in respired $CO_2$ is the result of a shift in the use of carbon substrates in the soil". Hence, likely better to refer to a shift in $\delta^{13}C$ or to "apparent fractionation" then to fractionation. Note also that they observed a strong change between the first days of incubations and subsequent days, and that there are some methodological aspects to consider when interpreting their data: high volume of soil (and water) compared to headspace, and complete flushing of the headspace with $CO_2$-free air (which implies that $CO_2$ dissolved in soil water remained and will re-equilibrate, this dissolved $CO_2$ has a different $\delta^{13}C$ value that the headspace $CO_2$, etc).**

**In short, this entire section at the moment lacks a solid empirical or theoretical basis to interpret differences in δ¹³C data observed to the influence of temperature or higher respiration rates. The same holds true for how some of the conclusions are expressed,**

*We appreciate this insight. We agree to revise the text in this section accordingly. First, we fully agree to refer to apparent fractionation rather than fractionation sensu stricto. Secondly, we propose to add $\delta^{15}N$ data in Table 1 to also show the trend in $\delta^{15}N$ signatures along elevation. Indeed, similar to $\delta^{13}C$ signatures, the $\delta^{15}N$ signatures decreased linearly with elevation. This indicates a more closed N cycling with increasing elevation. The results of $\delta^{13}C$ $\delta^{15}N$ signatures and $CO_2$ emission rates along elevation suggest a lower mineralisation at higher elevations. Similarly, the apparent $\delta^{13}C$ fractionation decreased linearly along the temperature gradient. This could indeed be due to a decrease in apparent fractionation and a shift in the use of carbon substrate. Finally, we agree to update some citations in this section. Therefore, we agree to revise the text in 4.1 (from line 452-463) as follows:*

*"The specific heterotrophic $CO_2$ emission decreased with increasing elevation, partly in response to effects of lower temperatures on microbial activity (Zimmermann et al., 2009). In support of the temperature effect on $CO_2$ respiration (Figure 3(b)), the apparent fractionation during SOC transformation to emitted $CO_2$ was also temperature-dependent. The emitted $CO_2$ at the warmer, lower elevations showed a higher apparent fractionation (and subsequently became relatively more depleted in $^{13}C$) than at the colder, higher elevations (Figure 3(c)). This observation may result from a shift in the use of carbon substrates along the temperature gradient or potentially $^{13}C$ discrimination during decomposition by soil micro-organisms (Andrews et al., 2000; Ehleringer et al., 2000; Natelhoffer and Fry, 1988). Indeed, the nitrogen stable isotope composition also indicated a decrease in $\delta^{15}N$ value with elevation (SI, Table 3). As such, based on the link between the nitrogen stable isotope composition of ecosystems and nitrogen cycling (Amundson et al., 2003; Boeckx et al., 2005), higher degrees of isotopic discrimination may indeed indicate a gradient in the rate of soil organic matter transformation processes likely including specific heterotrophic $CO_2$ emission. Our results imply that at higher elevations, even though SOC contents were high, microbial SOC decomposition was limited by lower temperatures (Zimmermann et al., 2009). In addition, low $CO_2$ emission at high elevations may as well partly be the result of the low soil pH values as those negatively affect microbial activity (Walse et al., 1998) and, thus, respiration (Figure 3(c), SI, Table 1). Further,*

*a low pH also facilitates the stabilisation of organic matter through complexation reactions with iron and aluminum ions, which become soluble at a low pH (Lützow et al., 2006)."*

**e.g. L 536-539: the statements made are not convincingly supported at the moment**

*Thank you for this comment. Following the revisions in section 4.1 above, we propose revise this statement to indicate the apparent fractionation.*

**L511-512: 'low soil moisture content limited microbial $CO_2$ respiration at high elevations: I do not see such lower moisture content anywhere in the data. Given the strong gradient in precipitation, I would expect to see rather the opposite ?**

*Thank you for this comment. The soil moisture content data indicated that the soil moisture content increased in the mid rainy season as compared to the start of the rainy season. Subsequently, $CO_2$ respiration increased in the same trend. Additionally, in the laboratory incubation, when soil moisture content was set uniform at 60% WFPS, we observed no difference in $CO_2$ respiration except when standardised per unit soil organic carbon, where it decreased linearly with elevation (suggesting an isolated temperature effect). These results indicate a boost of microbial activity probably due to the increase in soil moisture content in the wet season. Secondly, along the elevation gradient, soil moisture content tended to decrease with elevation in the mid rainy season (slope = -0.8, $R^2m$ = 0.25, SI, Figure 3). While this trend is not significant, the average WFPS was highest in the lowest elevation (57.2 % compared to the rest of the elevation from 1750-3000 (44.8 to 44.5 %, SI, Table 3), and the same trend was observed for $CO_2$ emission. For instance, $CO_2$ emission correlated strongly with water-filled pore space in the mid rainy season (P = 0.01, SI, Table 1). On the other hand, soil pH decreased linearly with elevation in the same trend as specific $CO_2$ emission. Subsequently, a significant correlation was observed between soil pH and $CO_2$ emission (SI, table 1).*
*Therefore, we agree to revise in the new manuscript to point at the correlations observed rather than the causal relationships.*

*Further, as already clarified about precipitation under comment L101 above, we do not have sufficient data within our sites elevations to determine any trend in precipitation*

**-L528-530: "we showed that…": rephrase this, limit to what you really unambiguously**

**demonstrate, relationships are not necessarily causal. For example, I do not see strong direct evidence that soil moisture or pH had a direct effect on soil respiration along your gradient.**

*While we appreciate that several variables change along elevation which makes it difficult to assess the effect of individual variables, we could point some relationships. Therefore, we propose to revise the language according to identify the correlative variables with $CO_2$ emission along the elevational gradient as explained under comment L511-512 above.*

**Minor / textual comments**

**-L20: insight into temperature sensitivity: insight into the temperature sensitivity:**

*Thank you, we agree to revise the sentence as suggested.*

**-L25 and further: temperature sensitivity: make it explicit that you are referring to $Q_{10}$ values here.**

*Thank you for this suggestion. We agree to revise as suggested*

**-L38: make it explicit that you refer to terrestrial primary production, not global (terrestrial + marine).**

*Thank you for this suggestion. We agree to rephrase the statement to clarify this.*

**-L61: delete "of the $CO_2$ respiration from soil"**

*Thank you for this suggestion. We agree to revise as suggested.*

**-L89: in the eastern slope: on the eastern slope.**

*Thank you for this correction.*

**-Equation 1: bit of an odd choice of symbols – (F, f, I, i)**

*Thank you for this comment. We agree to revise the symbol as in the above comment under line* 155-162, i.e.

$$"\delta^{13}C\text{-}CO_2 \approx \frac{[CO_2]_{final} * \delta^{13}C\text{-}CO_{2\,final} - CO_2]_{initial} * \delta^{13}C\text{-}CO_{2\,initial}}{[CO_2]_{final} - [CO_2]_{initial}} \qquad (1)"$$

'-L172: why "increment" ? I think this can be deleted.

*Thank you for the suggestion, we agree to delete the word.*

**References**

Amundson, R., Austin, A. T., Schuur, E. A., Yoo, K., Matzek, V., Kendall, C., Uebersax, A., Brenner, D., and Baisden, W. T.: Global patterns of the isotopic composition of soil and plant nitrogen, Global biogeochemical cycles, 17, https://doi.org/10.1029/2002GB001903, 2003.

Andrews, J. A., Matamala, R., Westover, K. M., and Schlesinger, W. H.: Temperature effects on the diversity of soil heterotrophs and the δ13C of soil-respired CO2, Soil Biology and Biochemistry, 32, 699-706, https://doi.org/10.1016/S0038-0717(99)00206-0, 2000.

Boeckx, P., Paulino, L., Oyarzún, C., Cleemput, O. v., and Godoy, R.: Soil δ15N patterns in old-growth forests of southern Chile as integrator for N-cycling, Isotopes in Environmental and Health Studies, 41, 249-259, https://doi.org/10.1080/10256010500230171, 2005.

Chung, H., Zak, D. R., Reich, P. B., and Ellsworth, D. S.: Plant species richness, elevated CO2, and atmospheric nitrogen deposition alter soil microbial community composition and function, Global Change Biology, 13, 980-989, https://doi.org/10.1111/j.1365-2486.2007.01313.x, 2007.

Davidson, E. A. and Janssens, I. A.: Temperature sensitivity of soil carbon decomposition and feedbacks to climate change, Nature, 440, 165-173, https://doi.org/10.1038/nature04514, 2006.

Denef, K., Bubenheim, H., Lenhart, K., Vermeulen, J., Van Cleemput, O., Boeckx, P., and Müller, C.: Community shifts and carbon translocation within metabolically-active rhizosphere microorganisms in grasslands under elevated CO 2, Biogeosciences, 4, 769-779, https://doi.org/10.5194/bg-4-769-2007, 2007.

Drijber, R. A., Doran, J. W., Parkhurst, A. M., and Lyon, D.: Changes in soil microbial community structure with tillage under long-term wheat-fallow management, Soil Biology and Biochemistry, 32, 1419-1430, https://doi.org/10.1016/S0038-0717(00)00060-2, 2000.

Ehleringer, J. R., Buchmann, N., and Flanagan, L. B.: Carbon isotope ratios in belowground carbon cycle processes, Ecological Applications, 10, 412-422, https://doi.org/10.1890/1051-0761(2000)010[0412:CIRIBC]2.0.CO;2, 2000.

Frostegård, Å. and Bååth, E.: The use of phospholipid fatty acid analysis to estimate bacterial and fungal biomass in soil, Biology and Fertility of soils, 22, 59-65, https://doi.org/10.1007/BF00384433, 1996.

Huygens, D., Roobroeck, D., Cosyn, L., Salazar, F., Godoy, R., and Boeckx, P.: Microbial nitrogen dynamics in south central Chilean agricultural and forest ecosystems located on an Andisol, Nutrient Cycling in Agroecosystems, 89, 175-187, https://doi.org/10.1007/s10705-010-9386-0, 2011.

Keeling, C. D.: The concentration and isotopic abundances of atmospheric carbon dioxide in rural areas, Geochimica et cosmochimica acta, 13, 322-334, https://doi.org/10.1016/0016-7037(58)90033-4, 1958.

Kroppenstedt, R.: Fatty acid and menaquinone analysis of actinomycetes and related organisms, Chemical methods in bacterial systematics, 173-199, 1985.

Natelhoffer, K. and Fry, B.: Controls on natural nitrogen-15 and carbon-13 abundances in forest soil organic matter, Soil Science Society of America Journal, 52, 1633-1640, https://doi.org/10.2136/sssaj1988.03615995005200060024x, 1988.

Phillips, D. L. and Gregg, J. W.: Uncertainty in source partitioning using stable isotopes, Oecologia, 127, 171-179, https://doi.org/10.1007/s004420000578, 2001.

Zelles, L.: Phospholipid fatty acid profiles in selected members of soil microbial communities, Chemosphere, 35, 275-294, https://doi.org/10.1016/S0045-6535(97)00155-0, 1997.

---

## Author Comment (AC2)

**Comment on bg-2022-37**

Anonymous Referee #2

Referee comment on "Temperature sensitivity of soil organic carbon respiration along the Rwenzori montane forests elevational transect in Uganda" by Joseph Okello et al., Biogeosciences Discuss., https://doi.org/10.5194/bg-2022-37-RC2, 2022

**It is my pleasure to read and review this manuscript written by Joseph Okello et al. I congratulate the authors on a very substantial piece of work, nicely written up, general nicely documented and discussed by the authors with novelty design and solid data. Indeed, it is interesting work. Indeed, the authors offer a manuscript that illustrates interesting findings supporting some hypotheses raised during the last years: first, that soil organic carbon respiration positively responses to soil temperature; second, that mineralization and depletion of readily available carbon in soil is also a regulator of soil organic carbon variation with the changing of soil physicochemical properties and microbial community-induced by climate warming over time. Overall I support publication of this work, yet I have some comments to be considered (moderate revisions).**

*Thank you very much for appreciating our work and equally for the careful review and insightful suggestions to further improve the manuscript. We are greatly humbled by your support!*

**Small comments are on Abstract /Conclusion to present the findings of the selected soil microbial community to be involved in the SOC respiration processes of Q10 models. And it is better to give a feedback to the findings. Also, SOC should be given an abbreviation in the beginning of the abstract.**

*We appreciate these suggestions. We agree to revise the manuscript to give feedback on microbial community that the microbial community structure was not affected along the climate gradient. Additionally, as suggested we shall abbreviate soil organic carbon as SOC in the abstract. It is a pity that we couldn`t discuss more on microbial community. We noted that*

*microbial community structure did not show significant effects with altitude nor $CO_2$ emission. We feel these results of microbial community structure along the climate gradient are important to include in the abstract. The result is consistent with several studies that found no effect of temperature on microbial community structure e.g. (Karhu et al., 2014; Nazaries et al., 2015; Wei et al., 2014).*

**Introduction: authors should give that the effect of soil microbial community on SOC during climate warming is not yet well established. Maybe this can be added to the introduction to better develop the current study. Not?**

*Thank you for this suggestion. Indeed, we agree to add a statement in the introduction about the controversies on the effect of soil microbial community on SOC in response to climate warming. i.e. "Several studies reported reduced microbial biomass in response to warming being linked to either depletion of labile carbon (Bradford et al., 2008; Knorr et al., 2005) or a decrease in carbon use efficiency (Allison et al., 2010; Tucker et al., 2013). However, other studies found no effect of climate warming on microbial community (Karhu et al., 2014; Nazaries et al., 2015; Wei et al., 2014). This means that the changes in soil $CO_2$ emissions upon warming result from alteration in the activity of native microbial community without altering microbial community structure."*

*Our study on microbial community along the microclimate gradient in montane forests is consistent with the latter findings.*

**References**

Allison, S. D., Wallenstein, M. D., and Bradford, M. A.: Soil-carbon response to warming dependent on microbial physiology, Nature Geoscience, 3, 336-340, https://doi.org/10.1038/ngeo846, 2010.

Bradford, M. A., Davies, C. A., Frey, S. D., Maddox, T. R., Melillo, J. M., Mohan, J. E., Reynolds, J. F., Treseder, K. K., and Wallenstein, M. D.: Thermal adaptation of soil microbial respiration to elevated temperature, Ecology letters, 11, 1316-1327, https://doi.org/10.1111/j.1461-0248.2008.01251.x, 2008.

Karhu, K., Auffret, M. D., Dungait, J. A., Hopkins, D. W., Prosser, J. I., Singh, B. K., Subke, J.-A., Wookey, P. A., Ågren, G. I., and Sebastia, M.-T.: Temperature sensitivity of soil respiration rates enhanced by microbial community response, Nature, 513, 81-84, https://doi.org/10.1038/nature13604, 2014.

Knorr, W., Prentice, I. C., House, J., and Holland, E.: Long-term sensitivity of soil carbon turnover to warming, Nature, 433, 298-301, https://doi.org/10.1038/nature03226, 2005.

Nazaries, L., Tottey, W., Robinson, L., Khachane, A., Al-Soud, W. A., Sørensen, S., and Singh, B. K.: Shifts in the microbial community structure explain the response of soil respiration to land-use change but not to climate warming, Soil Biology and Biochemistry, 89, 123-134, https://doi.org/10.1016/j.soilbio.2015.06.027, 2015.

Tucker, C. L., Bell, J., Pendall, E., and Ogle, K.: Does declining carbon-use efficiency explain thermal acclimation of soil respiration with warming?, Global Change Biology, 19, 252-263, https://doi.org/10.1111/gcb.12036, 2013.

Wei, H., Guenet, B., Vicca, S., Nunan, N., AbdElgawad, H., Pouteau, V., Shen, W., and Janssens, I. A.: Thermal acclimation of organic matter decomposition in an artificial forest soil is related to shifts in microbial community structure, Soil Biology and Biochemistry, 71, 1-12, https://doi.org/10.1016/j.soilbio.2014.01.003, 2014.

---

## Author Response (AR2)

**Report #1**

Suggestions for revision or reasons for rejection (will be published if the paper is accepted for final publication).

Overall, the authors addressed the main issues, but in a few cases I still feel some more clarity should be added, this should not be much work, see below for details.

Dear reviewer, thank you very much for appreciating our revisions of the manuscript based on your constructive comments. In addition, we are grateful to you for clearly pointing out the few areas that require more clarification in order to further improve the manuscript. In the following specific sections, we pay attention to address those concerns. The corrections are marked in the manuscript using tracked changes.

**Revised version, L 146-147:** "For measurement of  $\delta^{13}$ C values, the standard used was VPDB (Vienna Pee Dee Belemnite), while  $\delta^{15}$ N values were measured in reference to air": these are just the references values to define the scale – mention which actual standards you measured to calibrate the data.

**Author reply:** We realized that the description of the isotopic data was inadequate, and calibration information was missing, thank you for this remark. We reformulated as follows:

1) On the EA-IRMS measurements we reformulated as:

"The  $\delta^{13}$ C and  $\delta^{15}$ N values reported were normalized on the Vienna Pee Dee Belemnite (VPDB) and AIR scales using USGS90-milet flour (accepted  $\delta^{13}$ C and  $\delta^{15}$ N: -13.75 ± 0.06 ‰ vs. VPDB and +8.84 ‰ ± 0.17 vs. AIR, respectively) and USGS91-rice flour (accepted  $\delta^{13}$ C and  $\delta^{15}$ N: -28.28 ± 0.08 ‰ vs. VPDB and +1.78 ± 0.12 ‰ vs. AIR, respectively), High Organic Content Soil – 162517 (-26.27 ± 0.15 ‰ vs. VPDB and + 4.42 ± 0.29 ‰ vs. AIR, calibrated toward IAEA-CH-6 and IAEA-N-1 by Elemental Microanalysis Ltd) was used for quality analysis (QA). Standard deviation on replicate analyses of QA was better than 0.2 ‰ for both 13C and 15N, and deviation from certified value better than 0.2 and 0.3 ‰ for 13C and 15N respectively." (See line 118-125).

 Thanks to your remark we realized that normalization of the 13C measurement in CO2 using a laser based instrument was also not mentioned, therefore we added to that section:

"…analysed using Cavity Ring-Down Spectrometer, (G2113-I, CRDS CO2 analyser, Picarro, United States, normalization toward VPDB scale was done using a dilution in zero air of a 5 % CO2 ref gas calibrated by ISO ANALYTICAL toward IA-CO2 -3 ( $\delta^{13}C = -33.68$  ‰ vs. VPDB) traceable to NBS-19) at starting condition."(See line 162-163).

3) We also noted some discrepancies in the formulation of the equations 1) and 2) which were corrected.

**Original comment: section 2.3:** mention how you coped with removing a 45 mL gas sample from your incubation jars: was this volume replaced with air while taking these samples, if not how was the pressure difference accounted for in your measurements?

**Author reply:** We agree that this is needed. [...] While for laboratory incubation, headspace gas sampling for the determination of soil CO2 fluxes was done twice, directly at the beginning before fully closing the jar and at the end of the incubation period. For gas sampling a gas tight syringe (45 mL) was used, which was immediately gas-tightly closed after taking the gas sample. Gas samples were analyzed immediately using Cavity Ring-Down Spectrometer, (G2113-I, CRDS CO2 analyser, Picarro, UnitedStates). Thus, during incubation the pressure in the jar was not affected. Moreover, immediately after the second sampling, the jars were opened and only covered with parafilm to equilibrate until the following day.

**New comment:** This does not really answer the question – when you take 45 mL out of the incubation jar with a gas-tight syringe, you expand the volume hence the pressure decreases. Did you correct the data for the increase in volume, or did you replace the headspace volume while sampling.

Thank you for this inquiry. As we mentioned in the methodology section 2.3, the first sample was taken before closing the jar, so no problem of pressure change here. The second sample was taken at the end of the incubation in which during the withdrawal of the sample, the pressure will drop indeed, however, once the syringe is filled with 45mL of headspace, a valve on the syringe is closed so the lower pressure in the syringe will not induce an aspiration of lab air into the sample (actually the pressure in the syringe will immediately go back to atmospheric pressure by free movement of the plunger). As both Air  $(N_2/O_2)$  and CO2 can be considered as ideal gasses they will expand in the same manner, and the change in pressure during the sampling will thus not affect the concentration (the measurement by the CRDS is based on IR absorption of CO2 isotopologues present in a measuring cell at a fixed 'low' pressure and is thus only dependent on the concentration and not on the amount present in the syringe (the latter would be the case for a GC or a gas bench measurement). After the sample was taken, the jar was immediately reopened, and kept open (covered with a parafilm) until the next measurement moment. The lowering of the pressure in the headspace during the sampling could induce a certain outgassing of the soil however, seeing the very short time of sampling (couple of seconds) and the limited pressure drop (i.e. c.a. 4.5 %) we are convinced that this can easily be ignored.

**Original comment:** -L403-404: the data presented in Figure 4f show a surprisingly large difference in  $\delta^{13}$ C values, albeit with relatively high standard deviations. It would be good to provide statistics for this: for which elevations are these differences significant or not ? Again, I assume that for the high elevation sites, they are not statistically different – which should not be a surprise, as you have very organic soils here for which you would need to have a very high turnover rate to see any differences in  $\delta^{13}$ C of the SOC pool after 2 years. You could likely do some back-of-the envelope calculations here.

**Author reply:** Thank you for this insight. We agree with this comment. As mentioned earlier, L402-403 and L484, there were no significant differences between control and warmed samples

for a given elevation, while using linear mixed effect model regression analysis across entire elevational gradient allowed to show a trend between warmed and control treatments. As such we agree to clarify in the new version that "the remaining SOC tended to be more enriched in 13C in warmed than in control, and the SOC tended to be lower in warmed than in control treatment".

We agree to also add the statistics for SOC and  $\delta^{13}$ C in SI, Table 3 along with the slopes and intercepts of the linear mixed effect model regression fit in the new version of the manuscript.

**New comment:** From the statistics, it appears none of these differences are significant, although some close to. While this is acknowledged in the text, it is still phrased as "Finally, the  $\delta^{13}$ C values of the SOC showed that warmed soil became relatively more enriched in  $^{13}$ C as compared to control soil (Figure 4(f))." – that sort of statement suggests that your results are robust and significant, which they are not. Again, a quick back-of-the-envelope calculation could give you an indication of the turn over required to see the changes in d13C-SOC you report.

Thank you for this observation. We have now revised the statement to consistently state that there is only a trend but no significant difference.

i.e. "Finally, though not statistically significant, the  $\delta^{13}C$  values of the SOC showed a trend that warmed soil tended to become relatively more enriched in  $^{13}C$  as compared to control soil (Figure 4(f))."

Further, taking example of two elevation clusters at 1750-1850 and 2500-2600 m a.s.l, we applied the Rayleigh equation to estimate amount of SOC that should be respired for the observed changes in  $\delta^{13}C$  signature.

Fraction of SOC remaining =  $(\delta^{13}C \text{ warmed } + 1000) / (\delta^{13}C \text{ control } + 1000)^{(1/(alpha-1))}$

*Therefore, the fraction of SOC respired = 1-( fraction of SOC remaining).*

From this we saw that at 1750-1850 m a.s.l, up to ca. 44 % of SOC is needed to be respired in the two years of warming, while up to 81 % SOC is needed to be respired at 2500-2600 m a.s.l. The high SOC combined with low fractionation at the higher elevation (2500-2600 m a.sl), would require a high rate of SOC loss in order to observe changes  $\delta^{13}$ C. On the other hand, at the lower elevation where SOC content was lower and there is higher fractionation, a relatively smaller fraction is needed to be respired to observe the changes.

Nonetheless, we agree with you that these seem indeed very high turnover. The high turnover estimates may be a result of yet unexplained error.

We did not add these back-of-the envelope calculation to the MS as not to over-reach our available data.

**Original comment:** Throughout the manuscript, the terminology related to stable isotopes is really not OK. For example, the authors refer to "the 13C depletion factor" or "isotopic depletion factor" (L167) – that is not an accepted term in the literature, what you are referring to is termed fractionation (epsilon).

**New comment:** SI, Table 3 still mentions " $\delta^{13}$ C depletion factor »**

Thank you for this observation. We have corrected this to  ${}^{13}C$  isotopic fractionation during heterotrophic  $CO_2$  respiration (epsilon).

**Comments on Discussion, section 4.1.** I appreciate the changes made in the Discussion here; however given the non-significant differences I really would phrase this more cautiously. The text currently reads:

"Generally, after two years of in situ warming,  $\delta^{13}$ C values of SOC revealed a relative enrichment in 13C in warmed soil as compared to the control (Figure 4(f)). This is consistent with the observation of 13C depleted C losses during microbial CO2 respiration (Figure 3(c). The relative enrichment in 13C in warmed soil as compared to the control is likely due to enhanced mineralisation rates in the warmed soil. Higher mineralisation causes a change in 13C fractionation due to change in C substrate(following depletion of most labile C) and/or microbial discrimination against 13C during C transformation processes (Andrews et al., 2000; Ehleringer et al., 2000;Natelhoffer and Fry, 1988)."

-results on  $\delta^{13}$ C in warming vs control were not significant – I agree there is a tendency, but phrase it explicitly as such.

Thank you for your appreciation and for this insights. We have now rephrased the statement to only point out the depicted tendency or trend in  $\delta^{13}C$  in warming vs. control.